# The role of ion migration, octahedral tilt, and the A-site cation on the instability of $Cs_{1-x}FA_xPbI_3$

Weilun Li[1] ✉, Mengmeng Hao[2], Ardeshir Baktash [3], Lianzhou Wang [2,3] ✉ & Joanne Etheridge [1,4,5] ✉

Organic-inorganic hybrid perovskites are promising materials for the next generation photovoltaics and optoelectronics; however, their practical application has been hindered by poor structural stability mainly caused by ion migration and external stimuli. Understanding the mechanism(s) of ion migration and structure decomposition is thus critical. Here we observe the sequence of structural changes at the atomic level that precede structural decomposition in the technologically important $Cs_{1-x}FA_xPbI_3$ using ultralow dose transmission electron microscopy. We find that these changes differ, depending upon the A-site composition. Initially, there is a random loss of $FA^+$, complemented by the loss of $I^-$. The remaining $FA^+$ and $I^-$ ions then migrate, unit cell by unit cell, into an ordered and more stable phase with a $\sqrt{2} \times \sqrt{2}$ superstructure. Further ion loss is accompanied by A-site dependent octahedral tilt modes and associated tetragonal phases with different stabilities. These observations of the loss of $FA^+/I^-$ ion pairs, ion migration, octahedral tilt modes, and the role of the A-cation, provide insights into the atomic-scale structural mechanisms that drive and block ion loss and ion migration, opening pathways to inhibit ion loss, migration and improve structural stability.

Organic–inorganic hybrid perovskites (OIHPs) are promising materials for applications, including photovoltaics (PVs)[1–4], light-emitting diodes (LEDs)[5,6] and lasers[7,8]. They have a common $ABX_3$ structure (where A = organic cations, such as methylammonium ($MA^+$) and formamidinium ($FA^+$) or inorganic $Cs^+$; B = $Ge^{2+}$, $Sn^{2+}$, $Pb^{2+}$; and X = halogen). This flexible structure with A-site cations surrounded by the corner-sharing $[BX_6]^{4-}$ octahedral framework is thought to underpin the excellent power conversion efficiencies[9], long carrier diffusion lengths[10] and tuneable band gap[11] of OIHPs. However, several critical limitations on performance remain, including ion migration[12], current–voltage hysteresis[13,14] and material degradation in air and light[15,16]. Substantial efforts have been devoted to understanding correlations between the microstructure and physical properties of

OIHPs, thus finding strategies to improve their performance. However, an in-depth understanding of their structure and defect structure at the atomic level is still lacking. In particular, there are outstanding questions regarding how and where vacancies are formed and their role in ion migration, hysteresis, structural change, and ultimately decomposition.

Transmission electron microscopy (TEM) is a powerful tool for revealing the local atomic structure of materials, however, the pristine structure of OIHPs can be readily modified by the electron beam[17]. This presents challenges but also opportunities. Although extremely challenging, TEM data can be obtained from pristine structures if careful protocols for TEM specimen preparation, transfer and examination are applied, including ensuring the total electron dose lies well below

[1]School of Physics and Astronomy, Monash University, Clayton, VIC 3800, Australia. [2]Australian Institute for Bioengineering and Nanotechnology, The University of Queensland, St Lucia, QLD 4072, Australia. [3]School of Chemical Engineering, The University of Queensland, St Lucia, QLD 4072, Australia. [4]Monash Centre for Electron Microscopy, Monash University, Clayton, VIC 3800, Australia. [5]Department of Materials Science and Engineering, Monash University, Clayton, VIC 3800, Australia. ✉e-mail: weilun.li@monash.edu; l.wang@uq.edu.au; joanne.etheridge@monash.edu

measured damage thresholds. Under these controlled circumstances, there is an opportunity. While the physical mechanism of damage with light and electrons may be different, electron-induced damage can act as a proxy to provide insights into possible pathways for structural degradation. With careful control of additional electron dose above damage thresholds, sequences of structural change can be observed, revealing possible mechanisms for vacancy formation, ion migration, phase transitions and ultimately structural decomposition.

This approach has been applied variously to the pure $ABX_3$ systems, $MAPbI_3$, $MAPbBr_3$ and $FAPbI_3$, either in the form of bulk thin-film or quantum dots or nanocrystals. The structural response in these systems has been shown to be the same, irrespective of whether they are in quantum dot or bulk form[18–20]. In all cases, it has been observed that the final decomposition product is $BX_2$ (e.g., $PbI_2$ or $PbBr_2$)[18–24] or Pb[21,25,26]. However, the initial perovskite $ABX_3$ structures do not collapse directly into $BX_2$ but form an intermediate phase[18–20,23–25].

In the case of $FAPbI_3$, an intermediate phase was observed recently using moderately low dose atomic resolution scanning transmission electron microscopy (STEM) (dose ~200 e/Å²/s)[20]. This revealed a $\sqrt{2} \times \sqrt{2}$ ordered structure of $FA^+$ vacancies ($V^-_{FA}$) in the decomposition from cubic $FAPbI_3$ into $PbI_2$. This specific superstructure stabilized by ordered A-site vacancies has been proposed to explain the unusual regenerative properties of hybrid perovskite solar cells when degraded $MAPbI_3$ solar cells are post-treated with gaseous MAI[27].

In the case of $MAPbX_3$, intermediate phases were first detected from the appearance of additional (sometimes called forbidden) reflections in selected area diffraction (SAD) patterns (~1–2 e/Å²/ s)[18,23,24,28]. Different structural models were proposed for these intermediate phases, such as ordered halogen vacancies[18,24,28] and octahedral tilt or rotation[23], however, these models cannot be distinguished easily from SAD alone. Recently, further information was obtained for the specific case of $MAPbI_3$ using low-dose high-resolution TEM (HR-TEM) combined with a direct-detection electron counting camera (DDEC) which observed an intermediate phase of $MA_{0.5}PbI_3$ similar to that observed in $FAPbI_3$[19].

Despite these important observations, the mechanisms underpinning structural instability and ion migration remain unclear, including the mechanism by which ordered A-site vacancies are formed and whether there is any associated ordering of $I^-$ vacancies ($V^+_I$) and/or octahedral tilting. Furthermore, we need to understand what role, if any, the A-cation might play, as A-site engineering may provide an avenue for improving device stability[29]. We investigate these questions in the present paper for the technologically important mixed-cation perovskite $Cs_{1-x}FA_xPbI_3$. Such mixed-cation perovskites ($A_{1-x}A'_xBX_3$) have attracted great interest in photovoltaic applications due to their relatively good stability and charge transport properties[30–32]. The crystal structure of bulk and quantum dot $Cs_{1-x}FA_xPbI_3$ is the same, so either is suitable for this study[33–36]. We choose to examine quantum dots because they align consistently along a major zone axis, facilitating minimization of electron dose. We examine a high-quality synthesis that previously achieved a certified record power conversion efficiency of 16.6% at a composition of $Cs_{0.5}FA_{0.5}PbI_3$[37]. We also note that quantum dots have their own exciting applications in photo-active devices[37,38].

We first examine the pristine structures for pure $FAPbI_3$ and $Cs_{0.5}FA_{0.5}PbI_3$, and then the subsequent structural evolution at the atomic level, through ion-vacancy formation and ion migration, using ultra-low-dose HR-TEM with DDEC and low-dose annular dark-field STEM (STEM-ADF).

## Results

### Intermediate phase 1: A-cation and halogen vacancies ($V^-_A$ and $V^+_I$) and ordering

To identify unambiguously any atomic-level compositional variations, we first use low-dose STEM-ADF because the image intensity is directly related to the number and species of atoms in the atomic column (Fig. 1). The total electron dose was carefully minimized and measured by a direct electron detector to be 44 e/Å² per image (Supplementary note 1). This dose was the lowest we could use while still obtaining interpretable STEM-ADF images with atomic-level information (that is, sufficient signal-to-noise ratio (SNR)). However, we note that while this dose is several orders of magnitude lower than conventional STEM-ADF images, it still represents a significant dose for this class of materials, and we anticipate some electron-beam damage.

We consider first $FAPbI_3$, Fig. 1A–D. The pristine structure has previously been determined to be cubic (space group: $pm\bar{3}m$) using synchrotron x-ray diffraction (XRD)[37,39]. In the $\langle 001 \rangle$ zone axis atomic-resolution STEM-ADF image, the highest intensity maxima correspond to atomic columns comprising $Pb^{2+}/I^-$ with small local maxima in-between corresponding to $I^-$ columns (see intensity line scan along $\langle 100 \rangle$, Fig. 1C); the middle-intensity maxima correspond to $I^-$ columns and the lowest intensity maxima correspond to $FA^+$ columns due to less scattering to high angles from the organic molecule (Fig. 1B). In this image, it is evident that the image intensity at the $FA^+$ column positions alternates (high–low–high–low), suggesting a doubling of the original cubic unit cell. This is confirmed by the intensity line profile across the $FA^+$ column positions (Fig. 1D). Moreover, we also observe similar ordering of the intensity at the $Pb^{2+}/I^-$ columns and $I^-$ columns (Fig. 1C, D). These lower-intensity $FA^+$ columns and $I^-$ columns suggest the presence of vacancies. Moreover, an ordered pattern of both $FA^+$ and $I^-$ vacancies is evident, making a coordinated $\sqrt{2} \times \sqrt{2}$ superstructure of $V_{FA}$ and $V^+_I$ (Fig. 1I). We have confirmed this image interpretation using STEM-ADF image simulations. In particular, simulations show that the coordinated intensity modulations at the $FA^+$ and $Pb^{2+}/I^-$ and $I^-$ sites are not due to the effects of dynamical electron scattering. (supplementary figure 1).

We note in passing that despite the evident superstructure in the STEM images, the image SNR is too low to generate any detectable superlattice reflections in the corresponding Fourier transform (FT). Hence, we do not use FTs as a method to identify the minimum dose at which damage occurs (Supplementary Note 3). We also confirm that these vacancies and ordering are evident in the lowest dose raw data images and are not introduced by the post-filtering process (Supplementary Note 4).

Let us now consider the $Cs_{0.5}FA_{0.5}PbI_3$ (Fig. 1E–H). In this case, the A-site contains a mix of $FA^+$ and $Cs^+$ with the pristine structure previously being determined to be cubic (space group: $pm\bar{3}m$)[37]. As with $FAPbI_3$, we observe an alternating modulation of image intensity at all three atomic column sites ($Pb^{2+}/I^-$, $I^-$ and at the A-site, in this case, $FA^+$/ $Cs^+$ (Fig. 1E, F). Interestingly, the intensity line profiles indicate a smaller difference between high-intensity A-site columns and low-intensity A-site columns than for $FAPbI_3$, Fig. 1G, H. This suggests fewer $FA^+$/$Cs^+$ and $I^-$ vacancies have occurred in the mixed cation $Cs_{0.5}FA_{0.5}PbI_3$, compared with $FAPbI_3$ for the same electron dose. For $Cs_{0.5}FA_{0.5}PbI_3$, the A-site columns are nominally half occupied by $FA^+$ and half by $Cs^+$. We hypothesize that $FA^+$ vacancies occur more readily than $Cs^+$ vacancies (because $FA^+$, is known to readily break down into smaller molecules (such as $NH_3$ and $CH_2N$)[19,40]), so while $FA^+$ cations may be lost, $Cs^+$ cations remain on the A-site in sufficient numbers to generate intensity peaks in the ADF image. Hence, lower intensity peaks are still visible in the image at the vacancy-containing $FA^+$/$Cs^+$ columns, due to the remaining $Cs^+$ cations (Fig. 1J and Supplementary Note 5). Whereas for $FAPbI_3$, the lower intensity 'peaks' at the vacancy-containing A-site columns are barely visible or invisible.

To summarize the observations thus far, a low electron dose of 44 e/Å² applied with a scanned focussed electron probe, is sufficient to induce an ordered and coordinated $\sqrt{2} \times \sqrt{2}$ superstructure of A-site and $I^-$ vacancies in both $FAPbI_3$ and $Cs_{0.5}FA_{0.5}PbI_3$. Furthermore, image contrast is consistent with the A-site vacancies in $Cs_{0.5}FA_{0.5}PbI_3$ being predominantly $FA^+$ cations, rather than $Cs^+$.

## Formation mechanism−intermediate phase 1−FA$^+$/I$^-$ ion migration and ordering in FAPbI$_3$

A key question arises, namely, how do these ordered A-site and I$^-$ vacancy superstructures form from the initial undamaged cubic phase. We address this question by examining FAPbI$_3$ using an even lower dose imaging method, namely phase contrast HR-TEM combined with a DDEC. This technique can be performed with an order of magnitude lower dose than that of STEM-ADF and offers much better temporal resolution. However, the image contrast mechanism is different from STEM-ADF and the relationship between atomic column composition and image contrast is less direct. For this reason, we only consider images of FAPbI$_3$ where the A-site only comprises FA$^+$, so intensity variations at the A-site can be exclusively related to the occupancy of FA$^+$. With the DDEC, we obtained successive images, each with a dose of 1.5 e/Å$^2$ and then we summed sequences of these to provide images corresponding to a dose of our choice. This allows an identification of the dose at which structural changes begin and, critically, allows us to observe the subsequent structural changes, step-by-step, at the atomic level. We note that due to the different illumination conditions in STEM-ADF and in HR-TEM and the different way of estimating electron dose, the absolute dose may not be directly comparable and the critical dose for inducing structural changes may be different across the two techniques.

In the first instance, we sum six successive HR-TEM image frames, each acquired with a dose of ~1.5 e/Å$^2$, to form an image where the

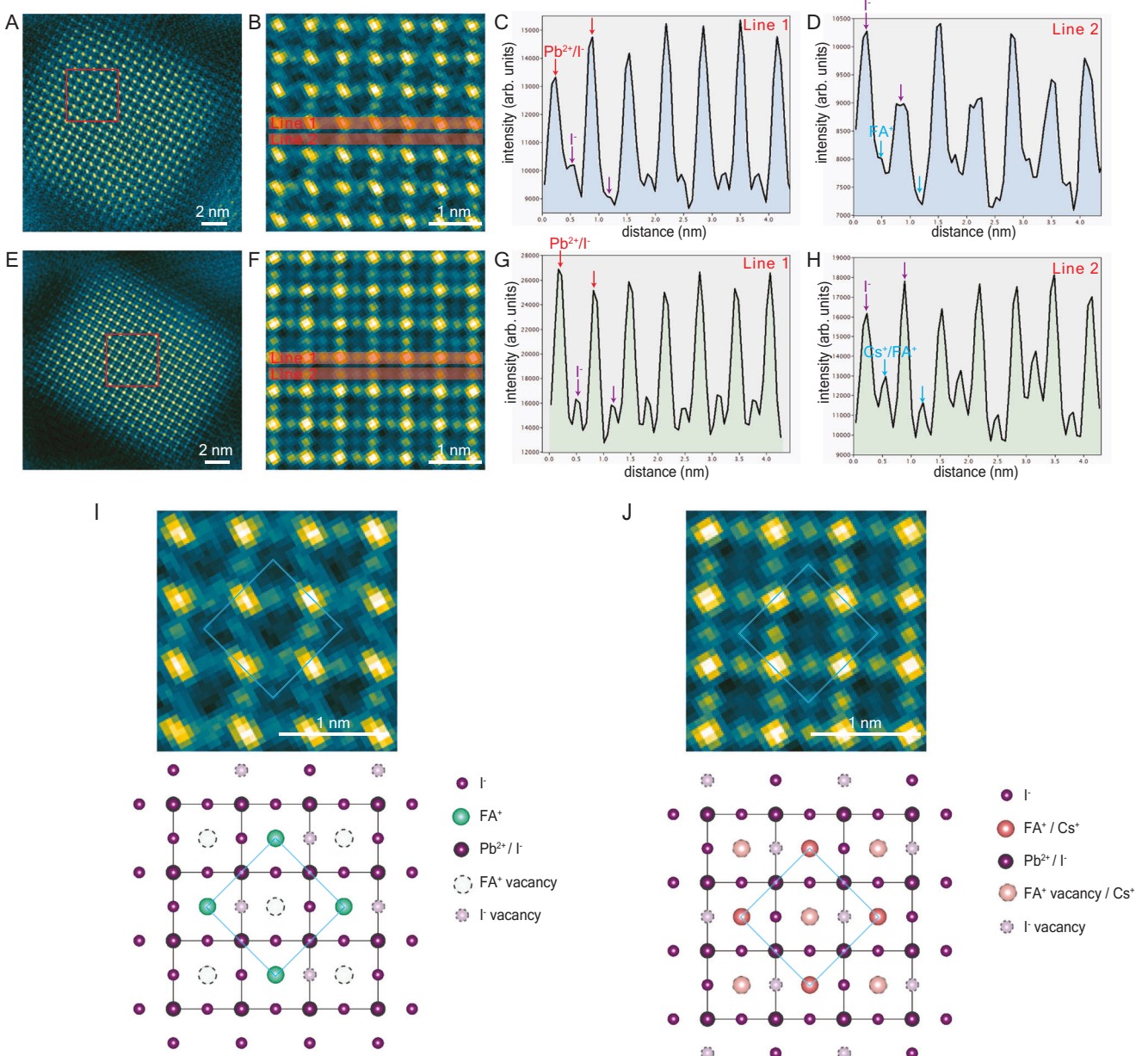

**Fig. 1 | A-site and I$^-$ vacancy ordering in FAPbI$_3$ and Cs$_{0.5}$FA$_{0.5}$PbI$_3$.** Low dose STEM-ADF images in the 〈001〉 zone axis. **A–D** FAPbI$_3$. **E–H** Cs$_{0.5}$FA$_{0.5}$PbI$_3$. **A**, **E** Lower-magnification images. Total dose is 44 e/Å$^2$. **B**, **F** Enlarged images of regions marked in (**A**, **E**). **C**, **G** Intensity line profiles integrated over Pb$^{2+}$/I$^-$ and I$^-$ columns as marked in region 1 in (**B**, **F**). **D**, **H** Intensity line profiles integrated over I$^-$ and FA$^+$ (or Cs$^+$/FA$^+$, in the case of Cs$_{0.5}$FA$_{0.5}$PbI$_3$) columns as marked in region 2 in (**B**, **F**). Arrows highlight atomic columns: red−Pb$^{2+}$/I$^-$; purple−I$^-$; blue−FA$^+$ (or Cs$^+$/FA$^+$). **I**, **J** schematic diagrams of vacancy ordering in FAPbI$_3$ and Cs$_{0.5}$FA$_{0.5}$PbI$_3$. Blue diamonds indicate the ordered remaining A-site cations. Low-dose STEM-ADF images are filtered by a combined Bragg filter and Butterworth filter to enhance image contrast. All structures and orderings present in the filtered images were also observed to exist in the raw images (details and raw images in Supplementary Note 4). Source data are provided as a Source Data file.

atomic structure can just be resolved with sufficient signal-to-noise ratio to permit quantitative measurements (Supplementary Notes 6 and 7). In the raw image, the $FA^+$ columns appear to have uniform image intensity and the corresponding FT is consistent with the pristine cubic perovskite structure, with no additional reflections. However, in the integrated column intensity map, some intensity variations are revealed (even though the precision of the column intensity analysis is affected by shot noise at such a low dose ($7.8\,e/Å^2$)). These variations are consistent with the presence of random $V^-_{FA}$ and $V^+_I$. Occasionally, there are even ordered vacancies in some local regions. Given the low dose, it could be that these vacancies are intrinsic to the pristine $FAPbI_3$, particularly if prepared with insufficient surface ligand (oleic acid). Or it could be that, even at this low dose, there are sufficient electrons to generate a few vacancies. Both explanations may apply. However, the number of vacancies that were observed at this stage (HR-TEM at $7.8\,e/Å^2$) is much less than in the first acquired STEM-ADF images taken with $44\,e/Å^2$ (Fig. 1). This suggests at least some of the vacancies observed in the first STEM-ADF images (Fig. 1) were induced by the electron beam, even though the STEM-ADF image was taken with the lowest achievable dose for STEM-ADF. We cannot know whether the vacancies incurred in the HR-TEM (taken at $7.8\,e/Å^2$) are intrinsic to the specimen or induced by the electron beam. We hypothesize that it is likely both are true.

To improve the SNR, we sum additional frames to generate an image corresponding to a total dose of $35\,e/Å^2$. The same observations apply as to the $7.8\,e/Å^2$ images but with greater clarity (Fig. 2I). At this stage, vacancies are still largely random and have not been ordered in a $\sqrt{2} \times \sqrt{2}$ superstructure, confirming there is a random initial loss of $FA^+$ and $I^-$ pairs. With a further increase in the total dose ($105\,e/Å^2$), the contrast in the image changes, most evident at $FA^+$ column positions (Fig. 2B). In the corresponding Fourier transform, additional $1/2,1/2,0_c$ and $1/2,3/2,0_c$ reflections are evident, inconsistent with the initial cubic structure with a space group $pm\bar{3}m$ (Fig. 2F). The integrated intensity map (Fig. 2J) shows significant variations in the $FA^+$ column intensity and in some regions, these have formed into an ordered pattern (e.g., region 1). With further continuous beam exposure (to $175\,e/Å^2$ and then $245\,e/Å^2$), $V^-_{FA}$ and its ordering can be observed directly in the HR-TEM images (Fig. 2C, D). In the FT, the number of forbidden reflections also increases, and the intensity of forbidden reflections becomes stronger (Fig. 2G, H). Most interestingly, the formation of a fully ordered $\sqrt{2} \times \sqrt{2}$ pattern of $V^-_{FA}$ vacancies is clearly visualized in the corresponding column intensity maps (Fig. 2K, L).

These HR-TEM image series reveal a step-by-step process of migration of $FA^+$ ions via A-site vacancies to form and an ordered $\sqrt{2} \times \sqrt{2}$ superstructure. For example, in region 1 from Fig. 2I, J, an ordered square of $V^-_{FA}$ is formed. This square of vacancies further diffuses by one unit cell (Fig. 2J and K), so the intensity is reversed (i.e., low-intensity $FA^+$ vacancy columns become high-intensity $FA^+$ occupied columns and vice versa). The same $V^-_{FA}$ diffusion process is observed in region 2 (Fig. 2K and L), resulting in a fully ordered pattern across the combined region. Our observations here suggest that the initial loss of $FA^+$ cations is random, and an ordered pattern is formed by the subsequent migration of $FA^+$ ions via $V^-_{FA}$. This demonstrates the mechanism of loss, migration, and ordering of $FA^+$ and is schematically illustrated in Fig. 2M–O.

In parallel with the above analysis of $FA^+$ vacancies, we performed a similar analysis of the intensity change of $Pb^{2+}/I^-$ and $I^-$ columns to study the presence of $I^-$ vacancies (Supplementary Note 8). Consistent with the observations from ADF-STEM, we find that the $I^-$ vacancies are correlated with the $FA^+$ vacancies. In addition, and significantly, we observe the same process of migration of iodine ions via $V^+_I$ to form a $\sqrt{2} \times \sqrt{2}\,V^+_I$ superstructure. This appears to occur in consort with the $FA^+$ vacancies and the formation of the $\sqrt{2} \times \sqrt{2}\,V^-_{FA}$ superstructure.

## Intermediate phases 2: A-site dependent octahedral tilting in $FAPbI_3$ and $Cs_{0.5}FA_{0.5}PbI_3$—initial observations

Thus far, we have observed the initial $FA^+/I^-$ vacancies and examined the associated $FA^+/I^-$ ion migration to form an ordered $\sqrt{2} \times \sqrt{2}\,V_{FA}$ and $V^+_I$ vacancy superlattice—the intermediate phase 1. We now investigate whether there are any subsequent structural changes with further exposure to the electron beam, before the established final decomposition to $PbI_2$. The first phase change, the ordered pattern of vacancies, was evident in ADF-STEM after the first scan at $44\,e/Å^2$ (and at lower doses in HR-TEM). We now examine a sequence of subsequent ADF-STEM images of $FAPbI_3$ and $Cs_{0.5}FA_{0.5}PbI_3$ taken with increasing electron dose.

In the case of $FAPbI_3$ (Fig. 3), the first scan (at ~$44\,e/Å^2$) exhibits the beginnings of a $\sqrt{2} \times \sqrt{2}\,V^-_{FA}$ and $V^+_I$ vacancy superstructure (just as we found in Fig. 1, confirmed by intensity line profiles in Supplementary Note 9). In the second scan (at ~$88\,e/Å^2$), very weak additional $1/2,3/2,0_c$ reflections (and their symmetry-equivalents, highlighted by red circles) appear in the FT (Fig. 3B, at $88\,e/Å^2$). After three scans (Fig. 3C, at $132\,e/Å^2$), these reflections are much stronger. Moreover, a distortion of the perovskite framework is clearly evident in the zoom-in image (Fig. 3M), consistent with octahedral tilting (see later). We expose for a further two scans and find there are no newly formed reflections nor any additional structural changes (Fig. 3D, E). We will call this, $FAPbI_3$—intermediate phase 2. (Note that this is not fully stoichiometric due to the loss of $FA^+/I^-$.)

In the case of $Cs_{0.5}FA_{0.5}PbI_3$ (Fig. 4), the first scan (at ~$44\,e/Å^2$) exhibits the start of a $\sqrt{2} \times \sqrt{2}\,V_{FA}$ and $V^+_I$ vacancy superstructure plus weak additional $1/2, -1/2, 0_c$ reflections (along the $\langle 1\text{–}10 \rangle$ direction) are just evident in the FT (Fig. 4A). These forbidden reflections correspond to an extra lattice frequency in the image in the $\langle 1\text{–}10 \rangle$ direction (perpendicular to the red arrows). We have carefully examined the first STEM-ADF scans of many $Cs_{0.5}FA_{0.5}PbI_3$ and find that the threshold dose at which such forbidden reflections can be first observed is in the range $44$–$132\,e/Å^2$ (as the first scan is at $44\,e/Å^2$, we cannot exclude the possibility that these forbidden reflections might appear below $44\,e/Å^2$). We suspect these small variations in the threshold dose for observing these additional $\pm 1/2, \pm 1/2, 0_c$ reflections are related to small composition variations in $Cs_{0.5}FA_{0.5}PbI_3$ (i.e., the $Cs^+/FA^+$ ratio).

In the next scan at $88\,e/Å^2$ (Fig S13 B), these $1/2, -1/2, 0_c$ reflections (along the $\langle 1\text{–}10 \rangle$ direction) disappear and a new set appears, $-1/2, 1/2, 0_c$, along the perpendicular direction, $\langle -110 \rangle$. By $750\,e/Å^2$, the structure appears to have stabilized with all $\pm 1/2, \pm 1/2, 0_c$ reflections present (Fig. 4C). We will call this, $Cs_{0.5}FA_{0.5}PbI_3$—intermediate phase 2. (Again, note that this is not fully stoichiometric due to the loss of $A^+/I^-$.)

Forbidden reflections then gradually disappear with a further increase in total dose, and the structure stabilizes into a square perovskite framework (Fig. 4D, E). By square, we mean $a = b$ and there is a $180°$ angle between octahedra, in contrast to the octahedral-tilted intermediate phase 2. (We cannot determine the third dimension nor the space group from this single projection.) It is surprising that instead of decomposing quickly into hexagonal $PbI_2$, the square perovskite framework remains, even up to a total dose of $3400\,e/Å^2$ (Supplementary Note 11). Although the FT of this square perovskite framework structure (Fig. 4E) is similar to that of the pristine cubic perovskite phase, a high density of vacancies ($V^-_{FA}$, $V^+_I$ and likely $V^-_{Cs}$) must be present. Moreover, throughout the progression from the 1st to the 2nd intermediate phase and then to the square perovskite framework structure in $Cs_{0.5}FA_{0.5}PbI_3$ (Fig. 4A–D), we notice the formation of local $PbI_2$ spherical clusters that fit coherently into the perovskite structure, as well as incoherent Pb clusters which quickly transform into coherent $PbI_2$ (Supplementary Note 12).

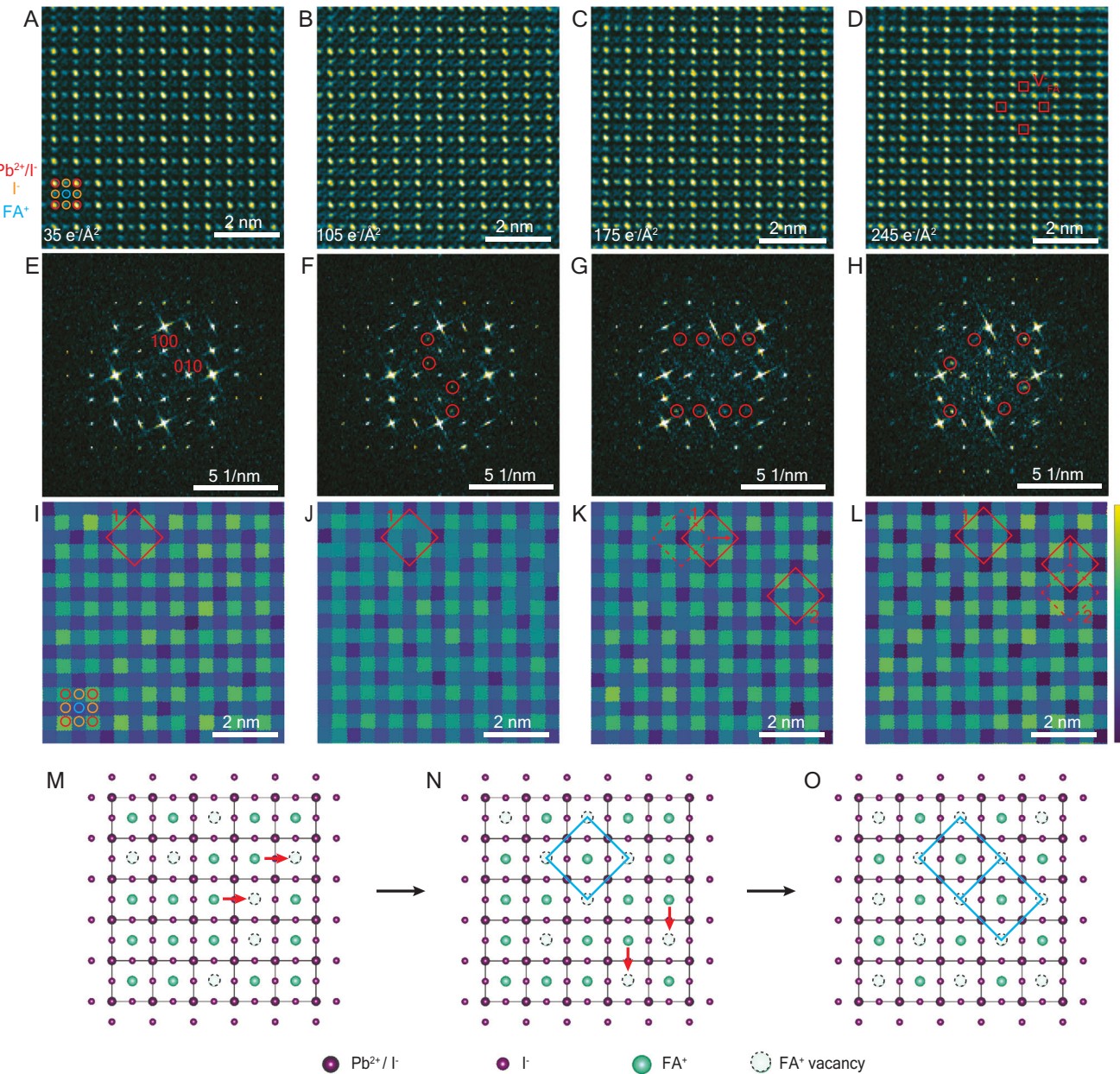

**Fig. 2 | Atomic-scale HR-TEM images of FAPbI₃ show the initial random FA⁺ vacancies and subsequent ordering via FA⁺ ion migration. A−D** HR-TEM images of FAPbI₃ with increasing electron beam exposure. Red circles correspond to the highest intensity Pb²⁺/I⁻ columns; orange circles correspond to I⁻ columns and blue circles correspond to the FA⁺ column. Red squares in (**D**) highlight ordered V⁻_FA. **E−H** FTs of images (**A−D**). Forbidden 1/2,1/2,0 and 1/2,3/2,0 reflections are marked by red circles. **I−L** Integrated column intensity maps based on HR-TEM

images in (**A−D**). Colour bar represents integrated intensity from 5000 to 10,000 in arb. units. Red solid diamonds highlight ordered V⁻_FA pattern in the image and red dashed diamonds highlight ordered V⁻_FA pattern in the previous image. **M−O** schematic diagrams illustrating the process of loss, migration and ordering of FA⁺ revealed in (**A−D**) and (**I−L**). Selected regions of ordered V⁻_FA are marked by blue diamonds. Red arrows indicate the migration of FA⁺. Black arrows between **M**, **N** and **N**, **O** indicate the loss and migration of ions with time series (or dose).

## Intermediate phases 2−Identification of A-site dependent octahedral tilt phases

The structures of intermediate phase 2 for FAPbI₃ and Cs₀.₅FA₀.₅PbI₃ are different, giving rise to different additional reflections as reported above. These two different phase 2 structures are determined here and shown in Fig. 5.

In the case of FAPbI₃, phase 2 can be attributed to an in-phase octahedral tilt ($a^0a^0c^+$ in Glazer notation), thus leading to $1/2,3/2,0_c$ forbidden reflections in the FT (Fig. 5A, B). We further find that this is consistent with a tetragonal $p4/mbm$ perovskite structure viewed in the [001] direction (Fig. 5C−E). We note that a similar tetragonal perovskite structure has been observed recently

in Cs₀.₀₅FA₀.₇₈MA₀.₁₇Pb(I₀.₈₃Br₀.₁₇)₃ by scanning electron diffraction[41]. The octahedral tilt in that system was proposed by the authors to be intrinsic to the pristine structure and to offer a stabilization mechanism for FA⁺-rich mixed-cation perovskites. Our observations here for FAPbI₃ are very different. In the case of FAPbI₃, this octahedral tilt phase is *not* present in the pristine structure. It is unequivocally electron beam induced and occurs at or below 88 e/Å² (at 300 kV in STEM mode).

In the mixed-cation Cs₀.₅FA₀.₅PbI₃, the ADF-STEM images of phase 2 show an elongation of the intensity maxima at the Pb²⁺/I⁻ column positions. This is found to result from an octahedral tilt mode ($a^+a^+c^0$ in Glazer notation). (note that we cannot determine the phase ± of

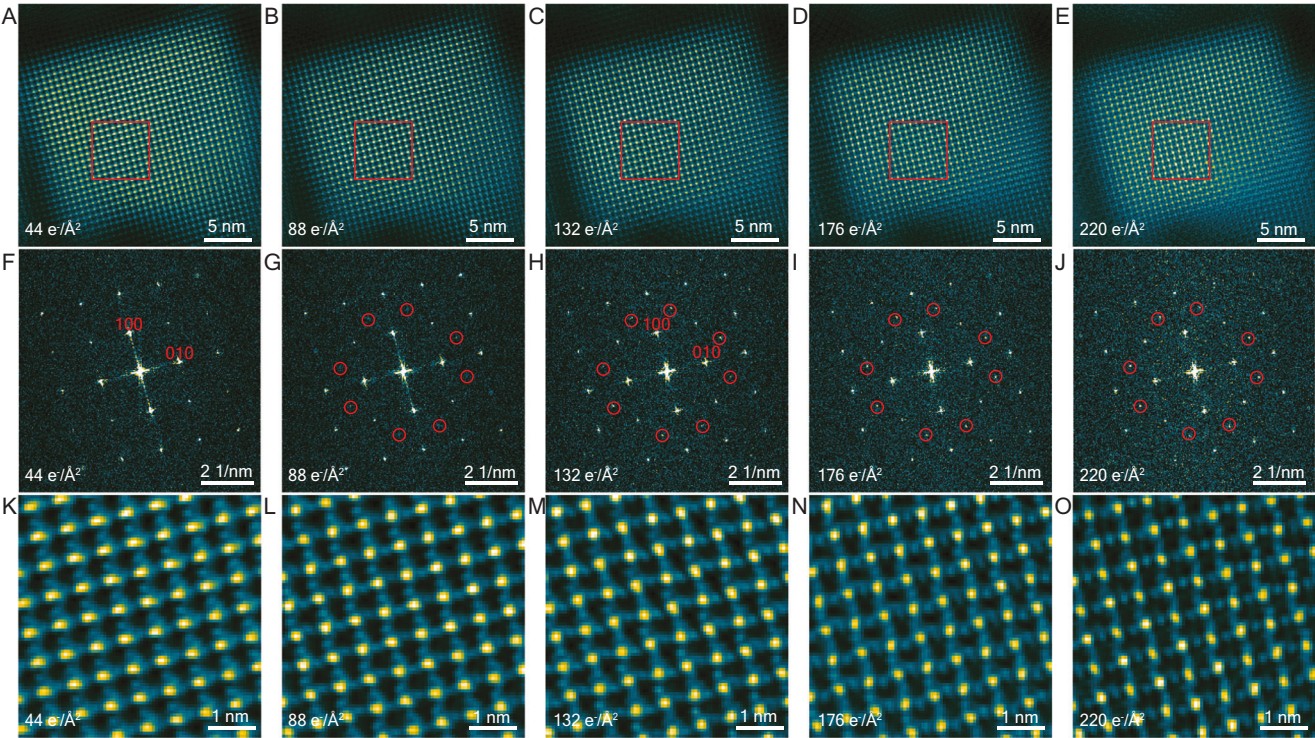

**Fig. 3 | Sequence of STEM-ADF images of FAPbI₃ shows the structural change of octahedral framework.** **A**–**E** STEM-ADF images with increasing total dose. **F**–**J** FTs of STEM-ADF images in (**A**–**E**). Reflections forbidden in the cubic structure are highlighted by red circles. **K**–**O** enlarged images of the regions marked in (**A**–**E**).

octahedral tilts from the projection). Specifically, this image is consistent with the [001] projection of a tetragonal $I4/mmm$ perovskite structure (Fig. 5F–J). This specific octahedral tilt mode results in strong forbidden reflections at $1/2,1/2,0_c$. We further notice that the $p4/mbm$ tetragonal phase observed for FAPbI₃ is also present in local regions of $Cs_{0.5}FA_{0.5}PbI_3$ (Supplementary Note 13). This suggests a local segregation of the A-cations to generate FA⁺-rich regions.

We note it has been reported in some parts of the literature that the pristine structure of $Cs_{1-x}FA_xPbI_3$ is consistent with the $I4/mmm$ tetragonal phase[42], same as in Fig. 5F. Our observations here of the 50/50 Cs/FA, $Cs_{0.5}FA_{0.5}PbI_3$ identify the pristine structure to be the $pm\bar{3}m$ cubic phase and the I4/mmm tetragonal phase to be unequivocally electron beam induced, occurring in the range <44 to 132 e⁻/Å² (at 300 kV in STEM mode).

## Discussion

This study provides direct insights, at the atomic-level, into the structural response to stimuli of the mixed cation perovskite $Cs_{1-x}FA_xPbI_3$ and its dependence on A-site composition (Fig. 6). The stimulus used here is an applied electron beam with extremely low current-density. This is used as a proxy with which to study the structural response to light, heat, and electric currents, and to understand at the atomic-level, how these stimuli cause the ion migration and structural degradation that are currently limiting device applications.

Point defects, such as vacancies, in photoactive perovskites are generally believed to be electronically benign, due to the observed long carrier diffusion lengths and low recombination rates[43]. However, while vacancies may be electronically benign, this study shows that they can be structurally toxic, being pivotal to ion migration and structural degradation and thereby undermining the potential of these materials for use in solar cell devices.

Even in the nominally pristine structure, occasional vacancy pairs are evident (Supplementary Fig. 8). Although low density, these

provide the initial space essential to permit ion movement and rearrangement. Once a stimulus is applied, additional vacancy pairs can form, further facilitating ion migration and local ordering. These are the first atomic-scale steps to device hysteresis and ultimately structural degradation.

We observe the loss of ion pairs, facilitating subsequent ion migration, unit cell by unit cell, leading sequentially to two intermediate phases, a vacancy-ordered phase, and then A-site dependent octahedral-tilt phases. These insights into the mechanisms of ion loss and migration suggest strategies for reducing ion migration and increasing structural stability which we discuss below.

1. Our first observation is that the structural change commences with a coincident and random loss of cation/anion (FA⁺ and I⁻) pairs, resulting in the formation of vacancies ($V^-_{FA}$ and $V^+_I$). That is, the loss of an FA⁺ cation will stimulate the loss of an I⁻ anion and vice versa. This applies to both FAPbI₃ and $Cs_{0.5}FA_{0.5}PbI_3$, however, we observed a slower rate of A-cation loss in the mixed cation compound because the bonding of FA⁺ with PbI₆ octahedra is weaker than that of Cs⁺. The initial loss of ions would result in a nominal formula of ($FA_{(1-x)}PbI_{(3-x)}$, $x = 0-0.5$) or ($Cs_{0.5}FA_{(0.5-x)}PbI_{(3-x)}$, $x = 0-0.25$). This loss of ions in pairs demonstrates that maintaining local charge neutrality is a dominant driving force within the crystal structure. This suggests that to enhance the structural stability of halide perovskites, it is crucial to suppress vacancies of either type. This in turn suggests that to engineer maximum structural stability of the photoactive phase, it is critical to introduce sources of both cations and ions (such as AX) that can limit or block both vacancy types, both during the initial perovskite material synthesis and for device fabrication purposes. A possible strategy for suppressing vacancy formation is the pinning of A cations and halide sites by enhancing the ionic bonding, for example, through the introduction of B site metal dopants, 2D lattices, or core–shell structures.

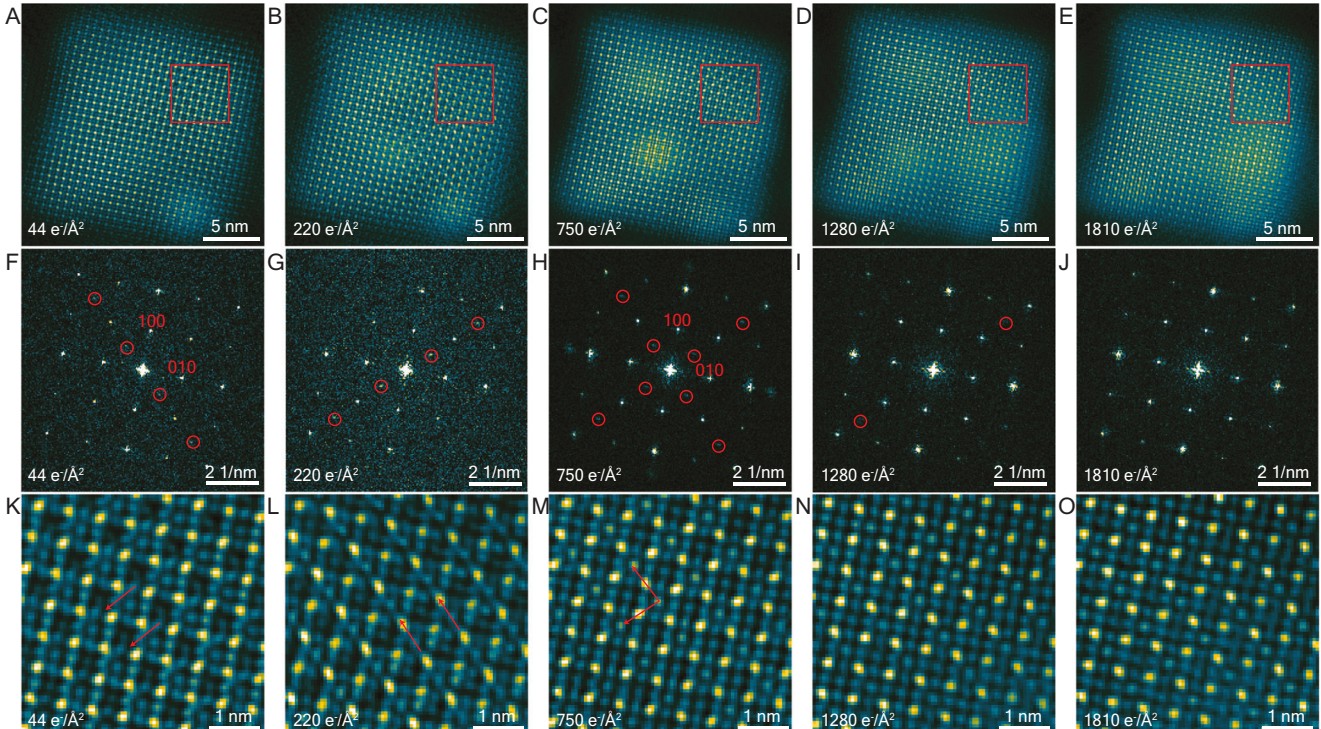

**Fig. 4 | Sequence of STEM-ADF images of $Cs_{0.5}FA_{0.5}PbI_3$ shows a structural change of octahedral framework that is different from $FAPbI_3$.** **A–E** STEM-ADF images with increasing total dose. **F–J** FTs of STEM-ADF images in (**A–E**). Reflections forbidden in the cubic structure are highlighted by red circles. (**K–O**) enlarged images of the regions marked in (**A–E**). Red arrows indicate extra lattice frequency that does not exist in the cubic perovskite structure. STEM-ADF images between 44 and 220 $e^-/Å^2$ are shown and discussed in Supplementary Note 10.

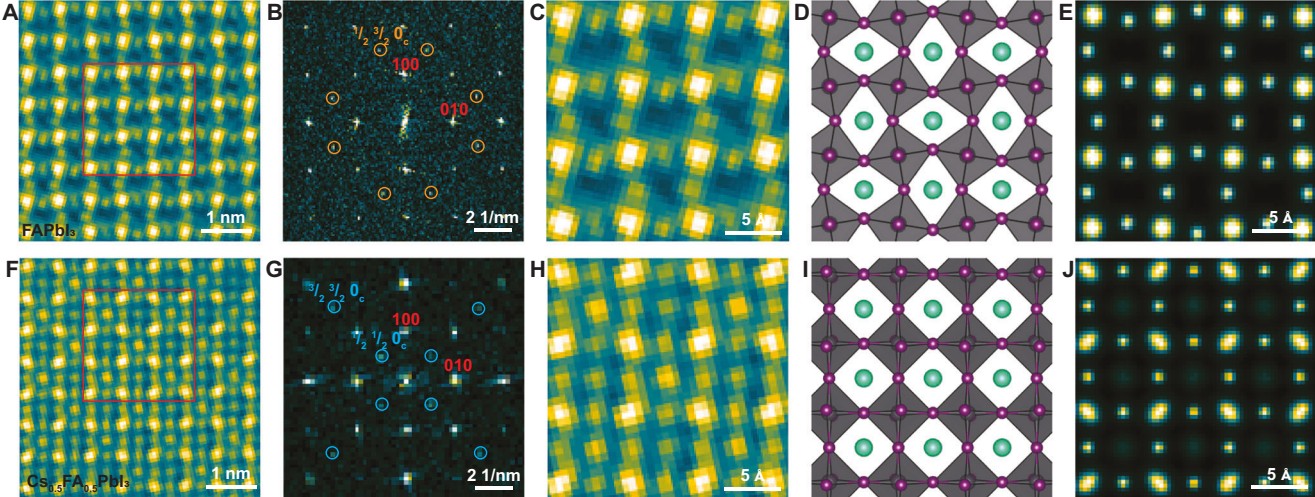

**Fig. 5 | Structural identification of intermediate phase 2 octahedral tilt modes in (A–E) $FAPbI_3$ and (F–J) $Cs_{0.5}FA_{0.5}PbI_3$.** **A, F** STEM-ADF image of fully established intermediate phase 2. **B–G** FTs from (**A, F**). Circles highlight reflections that are forbidden in the pristine cubic structure. **C, H** Unit cell structures of the region marked in (**A, F**). **D, I** The proposed structures showing different octahedral tilt modes for FAI-deficient $FAPbI_3$ ($a^0a^0c^+$) and $Cs_{0.5}FA_{0.5}PbI_3$ ($a^+a^+c^0$). Green, grey, and purple atoms represent $Cs^+/FA^+$, $Pb^{2+}$, and $I^-$ atomic columns, respectively. **E, J** STEM-ADF simulations based on the crystal structures proposed in (**D, I**).

2. Our second observation is that in the $FA^+/I^-$ deficient structure with randomly distributed vacancies, $FA^+/I^-$ ions demonstrate high mobility and promptly migrate toward an ordered super-structure. This shows that vacancies play a pivotal role in facilitating ion migration, indeed they may be a necessary condition for ion migration. This further emphasises the need to minimize vacancies in sample preparation, in order to passivate ion migration, improve structural stability and promote superior device performance. We have performed density function theory (DFT) calculations to better understand the A-site cation migration and the role of vacancy in the ion migration. A-site cation migration is not realistic in a perfect perovskite structure with fully occupied ions. DFT results show that the introduction of A-site vacancies will activate the migration of remaining A-site

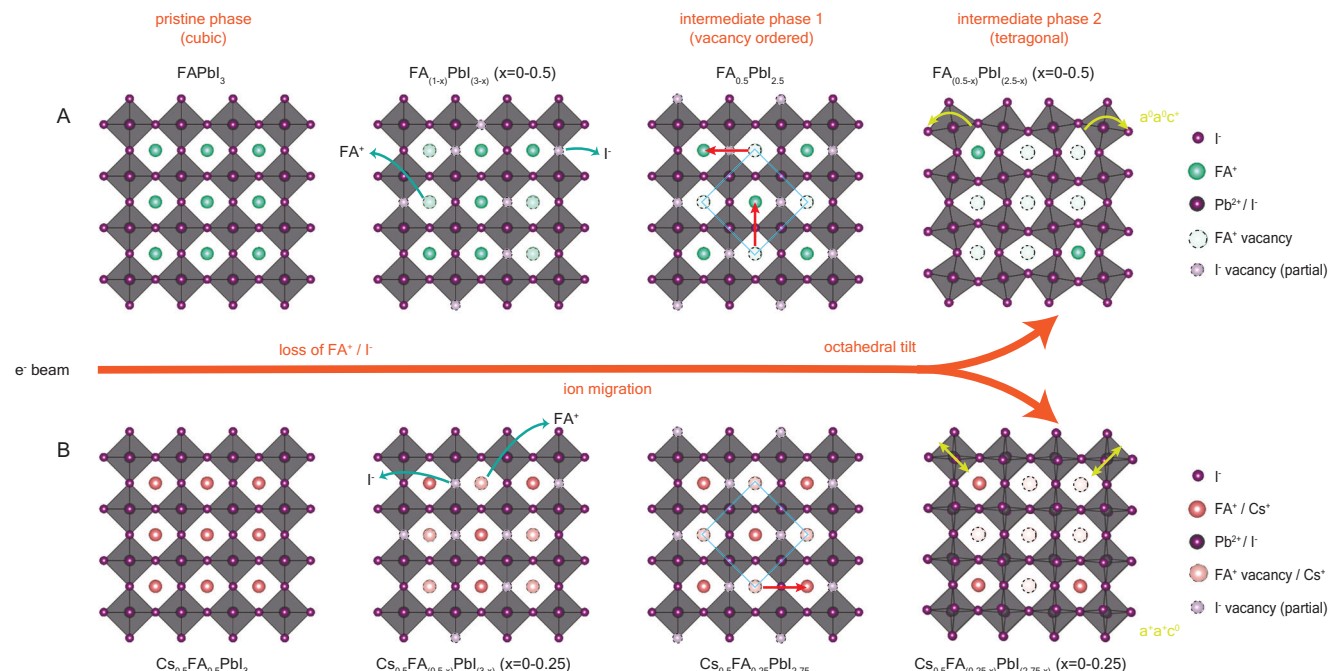

**Fig. 6 | Schematics of the observed ion migration mechanisms and associated phase changes of $Cs_{1-x}FA_xPbI_3$ under the electron beam. A** $FAPbI_3$. It involves an initial loss of $FA^+$ and $I^-$ (green arrows), followed by the formation of vacancy-ordered superstructure (intermediate phase 1, indicated by blue diamond) through ion migration (red arrows). With the further loss of ions, the pristine cubic phase transforms into a tetragonal phase through octahedral tilt (intermediate phase 2, indicated by yellow arrows). **B** $Cs_{0.5}FA_{0.5}PbI_3$. The loss and migration of ions are thought to be similar to $FAPbI_3$, however, the octahedral tilt mode for the intermediate phase 2 is different due to the alloying of $Cs^+$ at A-site.

cations (Supplementary Note 14). It suggests that vacancies provide a driving force for the subsequent re-ordering of cations through ion migration. We have also attempted to perform preliminary ab-initio molecular dynamics simulations to model the formation of the superstructure and its diffusion on a larger supercell scale. However, conclusive results have not been reached due to the complexity of the system, as discussed in (Supplementary Note 14).

3. Our third observation is that once vacancies have formed a $\sqrt{2} \times \sqrt{2}$ superstructure, a relatively stable intermediate phase (with the nominal formula of $FA_{0.5}PbI_{2.5}$ or $Cs_{0.5}FA_{0.25}PbI_{2.75}$), is established in which ion migration is substantially impeded, despite the presence of a significant number of $FA^+/I^-$ vacancies. This suggests the ordered configuration is energetically favourable, requiring a high activation energy to move an ion pair and break the local symmetry. It demonstrates how a well-ordered structure can discourage ion migration and suggests the design of well-ordered crystal structures, even those with ordered vacancies, might be one avenue for mitigating ion migration. Furthermore, it also indicates a potential atomic-level mechanism for how non-reversible ion migration and hysteresis might occur, namely, by driving ions into an ordered, energetically favourable superlattice from which larger energy is required to subsequently move them.

4. Our fourth observation is that, with the further loss of ions, the pristine perovskite Pb-I framework can no longer be maintained, and a second intermediate phase is formed through octahedral tilt resulting in a tetragonal phase transition. At this stage the nominal formula corresponds to ($FA_{(0.5-x)}PbI_{(2.5-x)}$, $x = 0$–0.5) or ($Cs_{0.5}FA_{0.25}PbI_{(2.75-x)}$, $x = 0$–0.25) before completely decomposing into $PbI_2$ (Supplementary Fig. 17). The octahedral tilt mode (or symmetry of the tetragonal perovskite phase) depends upon the type of A-site cation. The observed correlations between A-site cations (type and occupancy) and the Pb–I framework highlight

how the A-site can influence structural stability by tuning the octahedral tilt mode.

In summary, these observations reveal at the atomic scale the mechanisms by which ion migration occurs. This in turn suggests several strategies for designing the structure of perovskite photo-absorbers that will inhibit ion migration and promote structural stability, as follows:

- We find ion migration requires vacancies and these occur as cation/ion pairs. Chemical processes that block the formation of either vacancy type (cation or ion), will inhibit ion migration and enhance structural stability. Chemical processes and/or composition engineering that block both types (cation and ion), may block ion migration altogether.
- Well-ordered vacancy superstructures have higher stability and might be incorporated deliberately into the structure to discourage ion migration (for example, through low-temperature annealing in chemical processing).
- Structural stability can be enhanced through octahedral tilting, which might be induced through A-site composition engineering.

Since other stimuli such as high-intensity light, heat, or electrical fields may induce vacancy formation and ion migration in an analogous manner, these findings may provide fundamental instructions to guide the development of stable optoelectronic devices under various conditions.

## Methods

### Chemicals
All chemicals were purchased from Sigma Aldrich and used without purification unless otherwise noted. Caesium carbonate ($Cs_2CO_3$, 99.9%) lead iodide ($PbI_2$ 99.9985%, Alfa Aesar), oleic acid (OA, technical

grade 90%), oleylamine (OLA, technical grade 70%), 1-octadecene (ODE, technical grade 90%), toluene (anhydrous 99.8%), hexane (anhydrous, 95%), methyl acetate (MeOAc, anhydrous 99.5%), ethyl acetate (EtOAc, anhydrous 99.5%), lead(II) acetate trihydrate (Pb(OAc)$_2$·3H$_2$O, 99.999%), formamidine acetic acid salt (FA-acetate, ≥99%), oleylammonium iodide (OLA-I, ≥99%, Xi'an Polymer Light Technology Corp).

## Synthesis of CsPbI$_3$ QDs

Cs-oleate was obtained by dissolving 0.1 g of Cs$_2$CO$_3$ into 0.4 ml of OA and 10 ml of ODE, and the mixture was loaded into a 50-ml three-neck flask and stirred under vacuum for 30 min at 120 °C. After fully dissolving, the Cs-oleate in ODE was stored under nitrogen until it was used. PbI$_2$ (0.4 g), ODE (20 ml), OA (2 ml), and OLA (2 ml) were stirred in a 100-ml flask and degassed under vacuum at 120 °C for 1 h. The flask was then filled with N$_2$ and kept under constant N$_2$ flow. The temperature was increased to 170 °C, and then 3.4 ml of the Cs-oleate precursor was swiftly injected into the mixture. After 10 s, the reaction was quenched by immediate immersion of the flask into an ice bath. After cooling to room temperature, 30 ml of MeOAc was added, and the mixture was centrifuged at 6440 xg for 10 min. The resulting QD precipitate was dispersed well in 2 ml of hexane and was centrifuged again at 906×$g$ for 2 min to remove agglomerations. The concentration of the obtained QDs ink was further adjusted to 50 mg ml$^{-1}$ by adding the proper amount of hexane. Then the CsPbI$_3$ QDs ink was stored under nitrogen until use.

## Synthesis of FAPbI$_3$ QDs

Pb(acetate)$_2$·3H$_2$O (0.152 g), FA-acetate (0.157 g), ODE (16 ml) and OA (4 ml) were added in a 100-ml three-neck flask and dried under vacuum for 30 min at 40 °C. The mixture was then heated to 80 °C under an N$_2$ atmosphere, followed by an injection of OLA-I (0.474 g dissolved in 4 ml of toluene). After 30 s, the reaction mixture was cooled in the water bath. After cooling to room temperature, 20 ml of MeOAc was added, and the mixture was centrifuged at 6440×$g$ for 5 min. The resulting QD precipitate was dispersed in hexane and was centrifuged again at 1610×$g$ for 4 min to remove agglomerations and impurities. The concentration of the purified QDs ink was further adjusted to 50 mg ml$^{-1}$ by adding a proper amount of hexane. Then the QDs ink was stored under nitrogen until use.

## Synthesis of Cs$_{0.5}$FA$_{0.5}$PbI$_3$ QDs

Cs$_{0.5}$FA$_{0.5}$PbI$_3$ QDs were obtained by cation-exchange reactions: The stored CsPbI$_3$ QDs and FAPbI$_3$ QDs were mixed under N$_2$ atmosphere with a calculated volume ratio to guarantee the desired composition of the QDs. The ligand-assisted cation-exchange reaction was completed in 60 min at room temperature. The obtained QD ink was kept in an N$_2$-filled glovebox for an additional 12 h to guarantee the even distribution of surface ligands.

## Ligand density reduction

The surface ligand density of the obtained FAPbI$_3$ QDs and Cs$_{0.5}$FA$_{0.5}$PbI$_3$ QDs were further reduced by adding EtOAc (volume ratio of QD solution to EtOAc was 1:1) into the QD inks and centrifuged at 6440×$g$ for 5 min. The resulting QD precipitate was dispersed in hexane and was centrifuged again at 1610×$g$ for 4 min to remove agglomerations. The concentration of the QD inks was further adjusted to 50 mg ml$^{-1}$ by adding a proper amount of hexane. The purification process, including the mixing of QD inks with MeOAc or EtOAc, the centrifuge, and the dispersion of QDs in hexane was conducted in dry air with a relative humidity between 20% and 25%.

## TEM specimen preparation

QD solutions were stored in the glove box filled with dry N$_2$. Ultrathin carbon-coated Cu TEM grids were plasma-cleaned under H$_2$/O$_2$ for 30 s before use. A sample of QD solution was taken and immediately dropped onto the TEM grid and allowed to dry for 30 s. This whole sample preparation was conducted in the glove box (N$_2$). The prepared TEM specimen was transferred from the glove box to the TEM room, using a homemade stainless steel vacuum transfer unit. The total time of TEM specimen exposure in the atmosphere is <30 s.

## TEM characterization

STEM-ADF was carried out using an FEI Titan$^3$ 80-300 FEG-TEM equipped with probe and imaging spherical aberration correctors. All images were acquired at 300 kV, a 15 mrad probe-forming aperture, and 39–200 mrad detector collection angle. Electron dose was measured using an electron microscope pixel array detector (EMPAD) based on a measurement of 10,000 frames of the vacuum probe. HR-TEM was performed using a Thermo Fisher Scientific Spectra φ FEG-TEM equipped with a monochromator and probe and imaging C5 aberration correctors. HR-TEM images were acquired on a Gatan K3 camera in counting mode at 75 fps. All TEM experiments followed a strict "shoot blind" protocol whereby a fresh region of the specimen is first exposed to the electron beam at the start of data acquisition and only exposed for the duration of data acquisition. In particular, no electron dose was applied to the material for tilting to a zone axis or adjusting imaging parameters. Images were post-filtered with a combined Bragg filter and a Butterworth filter and corresponding raw images are given in the supplementary material.

## STEM simulations

STEM-ADF images were simulated using a GPU-enhanced frozen-phonon multislice code (µSTEM). The simulations employed supercells by tiling perovskite unit cells by 8 × 8 (supercells in size 5 × 5–10 × 10 nm), combined with 1024 × 1024 pixels to ensure accuracy. Experimental conditions were used as the parameters for the calculation of the STEM images. 30 frozen phonon passes were calculated.

## DFT calculations

DFT as implemented in the Vienna Ab Initio Package (VASP)[44] is used to study the Cs$_x$FA$_{1-x}$PbI$_3$ perovskite structure. The GGA-PB[45] functional is considered for all the calculations. To minimize the effect of periodic boundary conditions on atomic interactions, a supercell consisting of 8 unit cells with 96 atoms (12.703 Å × 12.703 Å × 12.703 Å) for FAPbI$_3$ and a supercell with 40 atoms (12.564 Å × 12.564 Å × 12.564 Å) for CsPbI$_3$ are considered for this study. A $k$-point of 1 × 1 × 1 at the Γ-point is applied for all the calculations. To compare the migration energy barrier (NEB) of FA and Cs ions in the structure of FAPbI$_3$ and CsPbI$_3$, nudge elastic band calculations with a force tolerance of 0.03 eV Å$^{-1}$ and an energy cutoff of 520 eV are carried out. To make the initial and final structures, one FA and I and similarly one Cs and I are removed from FAPbI$_3$ and CsPbI$_3$ structures (FA$_{0.875}$PbI$_{2.875}$ and Cs$_{0.875}$PbI$_{2.875}$) and DFT ground state optimization is used to converge the initial and final migration points. The energy-optimized structures are used to generate six intimidating images for NEB calculations.

## Ab-initio molecular dynamics (MD) simulation

The samples that were used for NEB calculations are also being used for Born−Oppenheimer molecular dynamics simulations. Ab initio MD simulations were performed using the CP2K/Quickstep package[46]. The Perdew−Burke−Ernzerhof (PBE) generalized gradient approximation (GGA)[45] was selected for the DFT exchange-correlation functional. To correct for van der Waals interactions the DFT-D3[47] method was used. Pseudopotentials of Goedecker, Teter, and Hutter (GTH)[48] were employed and the DZVP-MOLOPT-SR-GTH[49] was selected as basis set. This is a Gaussian and plane-wave (GPW)[50] basis and a cutoff energy of 280 Ry is selected. A 1 × 1 × 1 $k$-point mesh (Γ point) was used in all calculations. For ab initio MD simulations, the equations of motions were integrated using a velocity Verlet algorithm with a time step of 1 fs.

## Data availability

All relevant data generated in this study are provided in the paper and its Supplementary Information files. Source data has also been deposited in Figshare under the accession link https://doi.org/10.6084/m9.figshare.24615702[51]. Source data are provided with this paper.

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

## Acknowledgements

Financial support from the Australian Research Council (ARC) is appreciated. J.E. acknowledges ARC Discovery Project DP200103070 and ARC Laureate Fellowship FL220100202. L.W. acknowledges ARC Laureate Fellowship FL190100139. The authors acknowledge the use of facilities within the Monash Centre for Electron Microscopy, a node of Microscopy Australia. The Thermo Fisher Scientific Spectra φTEM was funded by ARC LE170100118 and the FEI Titan³ 80-300 FEG-TEM was funded by ARC LE0454166. The authors acknowledge access to computational resources at the NCI National Facility and Pawsey Supercomputing Centre through the National Computational Merit Allocation Scheme supported by the Australian Government. The authors also acknowledge support from the Queensland Cyber Infrastructure Foundation (QCIF) and the University of Queensland Research Computing Center (UQ RCC).

## Author contributions

W.L., L.W., and J.E. conceived and designed the project. J.E. and L.W. supervised the project. W.L. performed electron microscopy experiments, simulations and analysed the data. M.H. synthesized the perovskite quantum dots. A.B. performed the density function theory calculations. W.L. and J.E. prepared the manuscript. All authors contributed to the discussion of the results and revision of the manuscript.

## Competing interests

The authors declare no competing interests.
