## [Peer Review File · Nature Communications]

REVIEWER COMMENTS

Reviewer #1 (Remarks to the Author):

The ultra-low dose HR-TEM experiments carried out by the authors have been carefully executed and significant effort has been put into drawing conclusions. However, it is not clear after reading the manuscript as to how these findings help address the instability issues of hybrid perovskites. Authors start the Abstract with an emphasis on the bulk perovskites and then discuss their results on 'perovskite quantum dots'. It is not clear how insights on quantum dots can advance the field of bulk perovskites. In fact, this remains the theme across the Introduction section as well, with the bulk perovskites receiving most of the attention, but with the narrative switching to perovskite quantum dots toward the end.

Also, the Introduction section is very long and can be shortened.

The Results section appears very technical and specifically focused and tuned toward a 'nanostructure/electron microscopy' audience. It is not clear how this paper will serve the broader perovskite community. In its current form, this paper appears to be a better match for a specialized journal and not fit for Nature Communications that has a very broad scope.

When the authors draw conclusions about vacancy formation as a result of the electron dose, it is not clear how this relates to device behavior. In general, perovskites are known to be intrinsically defect tolerant. Is the defect density introduced by the electron dosage sufficient to create noticeable device performance loss? Are these defect densities and defect types a concern for the technologies expected from these semiconductors?

In general, it is not clear how the strategies suggested can be generalized to the broader and more promising field of bulk perovskites. Authors could have additionally carried out these experiments for a bulk perovskite as well. It is clear that perovskite quantum dots, as such, are not as compelling from a technological perspective. This study could also have benefited from theoretical modeling of these defects pinpointing the location of the resulting defect states in the band structure.

This is a robust analysis on using low-dose TEM for creating and observing structural changes at the atomic scale in perovskite quantum dots, however, it is not clear how this can technologically advance the field of bulk perovskites. Therefore, I cannot suggest publication of this work in Nature Communications.

Reviewer #2 (Remarks to the Author):

In this study, Li and coauthors investigate the instability of $\text{Cs}_{1-x}\text{FAxPbI}_3$ and clarify the role of ion migration, octahedral tilt and the A-site cation via low-dose TEM. They directly image the detailed structural evolutions of $\text{Cs}_{1-x}\text{FAxPbI}_3$ under the electron beam irradiation including the initial random loss of vacancies, forming superstructures and inducing the octahedral tilt. Overall, this work provides insights into the atomic-scale structural mechanisms of ion loss and migration and suggests strategies for reducing ion migration and increasing structural stability. The manuscript is well organized and written. Several issues need to be addressed before publication, which are detailed below.

My first and foremost concern is the influence of post-filtering. In the raw image, there is no additional reflections while, in the filtered images, additional reflections and the intensity modulation are observed. Why these filters can produce additional reflections? Besides, the ABSF and Winer filter images seem to match better with the raw image and no superstructure reflections are observed in the corresponding FFT pattern. However, the vacancies there seem not ordered. Thus, how to identify that the observed intensity-modulation is the pristine feature in the raw images rather than induced by the Bragg filter and butterworth filter? I suggest the author analyzes the intensity modulation from the images containing additional

reflections in the raw images (just like STEM image in Fig. 4a).

Please provide the raw image and methods of filtering for each image.

How to identify the type of atoms in phase-contrast HRTEM images?

Could the author provide a more quantitative analysis on estimating the vacancy content based on the intensity of the STEM images for Fig. 1?

Fig. 1b and 1f, the region of lineprofiles is not clearly indicated.

FFT patterns in Fig. 1 is too small and should be removed to SI. Moreover, FFT patterns in Fig.2-5 can be further cropped to make the spots clearer.

Reviewer #3 (Remarks to the Author):

The structural degradation of organic-inorganic hybrid perovskite solar cells is a major obstacle in their commercialization. To investigate the degradation mechanism, the authors used transmission electron microscopy-based techniques to explore the effects of ion migration, octahedral tilt, and the A-site cation on $\text{Cs}_{1-x}\text{FA}_x\text{PbI}_3$ instability. The authors made several notable novel findings, including the atomic-scale dynamic observation of superstructure formation and distinct degradation pathways for FAPbI_3 and $\text{Cs}_{1-x}\text{FA}_x\text{PbI}_3$, which significantly advance the understanding of its degradation mechanism. However, there are some comments that the authors should take into account before publication:

1. Although the authors mainly focused on investigating the initial intermediate phases, it is essential to consider the evidence for the transition from the intermediate phase to the final PbI_2 phase, which is well known to result from the decomposition of ABi_3 .
2. The authors observed an abnormal phenomenon of vacancy-induced superstructure formation, but the proposed discussion of ordered cation diffusion needs proper simulations for better understanding.
3. The authors should clarify whether the random vacancies observed in Figure 1 are due to beam irradiation or sample preparation.
4. The authors claim that charge neutralization plays a crucial role in cation vacancy formation, which suggests that the number of vacancies for A cations and I⁻ should be similar. It would be helpful if the authors could provide quantitative analysis from the images to support this claim.
5. The authors named intermediate as phase 1 and phase 2, and it would be useful to discuss the corresponding non-stoichiometric chemical formula based on the observations.

RESPONSE TO REVIEWERS' COMMENTS

Manuscript ID: NCOMMS-23-20206

Title: "Instability of Cs_{1-x}FAPbI₃ quantum dots – the role of ion migration, octahedral tilt and the A-site cation"

Author(s): Li, Weilun; Hao, Mengmeng; Wang, Lianzhou; Etheridge, Joanne

We thank the reviewers for their time and helpful comments. We have endeavoured to address these comments in the revised manuscript, as detailed below. (Reviewers comments in **black bold**, our reply in **blue**, changes to the manuscript in **red**. Page and line number refers to the original manuscript)

Reviewer #1 (Remarks to the Author):

The ultra-low dose HR-TEM experiments carried out by the authors have been carefully executed and significant effort has been put into drawing conclusions. However, it is not clear after reading the manuscript as to how these findings help address the instability issues of hybrid perovskites. Authors start the Abstract with an emphasis on the bulk perovskites and then discuss their results on 'perovskite quantum dots'. It is not clear how insights on quantum dots can advance the field of bulk perovskites. In fact, this remains the theme across the Introduction section as well, with the bulk perovskites receiving most of the attention, but with the narrative switching to perovskite quantum dots toward the end.

Also, the Introduction section is very long and can be shortened.

The Results section appears very technical and specifically focused and tuned toward a 'nanostructure/electron microscopy' audience. It is not clear how this paper will serve the broader perovskite community. In its current form, this paper appears to be a better match for a specialized journal and not fit for Nature Communications that has a very broad scope.

When the authors draw conclusions about vacancy formation as a result of the electron dose, it is not clear how this relates to device behavior. In general, perovskites are known to be intrinsically defect tolerant. Is the defect density introduced by the electron dosage sufficient to create noticeable device performance loss? Are these defect densities and defect types a concern for the technologies expected from these semiconductors?

In general, it is not clear how the strategies suggested can be generalized to the broader and more promising field of bulk perovskites. Authors could have additionally carried out these experiments for a bulk perovskite as well. It is clear that perovskite quantum dots, as such, are not as compelling from a technological perspective. This study could also have benefited from theoretical modeling of these defects pinpointing the location of the resulting defect states in the band structure.

This is a robust analysis on using low-dose TEM for creating and observing structural changes at the atomic scale in perovskite quantum dots, however, it is not clear how this can technologically advance the field of bulk perovskites. Therefore, I cannot suggest publication of this work in Nature Communications.

We appreciate the reviewer's time and comments. We address each point below.

Point 1. "...it is not clear after reading the manuscript as to how these findings help address the instability issues of hybrid perovskites."

Fundamentally, the electron beam, even when used with ultralow doses as in our study, is a source of external stimulus, passing energy on to the perovskites to trigger the ions to overcome the diffusion energy barrier, leading to ion migration and subsequent structure changes. This behaviour is similar to other stimuli (e.g. light, heat) induced structural degradation in the bulk-film based devices. As commented by the reviewer 3, the findings will help better understand the degradation mechanisms of the perovskites with similar compositions which have been used in a range of optoelectronic devices.

We have endeavoured to clarify and further emphasise this by adding the following text to the conclusion:

Page 9, Line 18:

This study provides direct insights, at the atomic-level, into the structural response to radiation stimuli of the mixed cation perovskite $\text{Cs}_{1-x}\text{FA}_x\text{PbI}_3$ quantum dots and its dependence on A-site composition (Fig. 6). The stimulus used here is an applied electron beam with extremely low current-density. This is used as a proxy with which to study the structural response to light, heat and electric currents, and to understand *at the atomic-level*, how these stimuli cause the ion migration and structural degradation that is currently limiting device applications.

and Page 10, Line 33:

In summary, these observations reveal at the atomic scale the mechanisms by which ion migration occurs. This in turn suggests several strategies for designing the structure of perovskite photoabsorbers that will inhibit ion migration and promote structural stability, as follows:

- We find ion migration requires vacancies and these occur as cation/ion pairs. Chemical processes that block the formation of either vacancy type (cation or ion), will inhibit ion migration and enhance structural stability. Chemical processes and/or composition engineering that block both types (cation *and* ion), may block ion migration altogether.
- Well-ordered vacancy superstructures have higher stability and might be incorporated deliberately into the structure to discourage ion migration (for example, through low temperature annealing in chemical processing).
- Structural stability can be enhanced through octahedral tilting, which might be induced through A-site composition engineering.

Point 2: "It is not clear how insights on quantum dots can advance the field of bulk perovskites."

The perovskite nanocrystals, or quantum dots, have the same crystal structure as their bulk counterparts, so the observations made here are equally applicable to bulk perovskites and quantum dots. To clarify this, we have added the following text, in red, and moved the position of the text in green:

Page 3, Line 4:

We investigate these questions in the present paper for the technologically important mixed-cation perovskite $\text{Cs}_{1-x}\text{FA}_x\text{PbI}_3$ ~~quantum dot system~~. Such mixed-cation perovskites ($\text{A}_{1-x}\text{A}'_x\text{BX}_3$) have attracted great interest for photovoltaic applications due to their relatively good stability and charge transport properties^{30–32}. The crystal structure of bulk and quantum dot $\text{Cs}_{1-x}\text{FA}_x\text{PbI}_3$ is the same, so either is suitable for this study^{33–36}. We choose to examine quantum dots because they align consistently along a major zone-axis, facilitating minimisation of electron dose. We examine a high quality synthesis which previously achieved a certified record power conversion efficiency of 16.6% at a composition of $\text{Cs}_{0.5}\text{FA}_{0.5}\text{PbI}_3$ ³⁷.

To improve clarity further, the revised manuscript has been modified so it only refers specifically to ‘quantum dots’ where it is relevant, that is, wherever a statement is unique to quantum dots and not applicable to bulk $\text{Cs}_{1-x}\text{FA}_x\text{PbI}_3$.

“Quantum dot” and “QD” have been removed accordingly from the main manuscript and the SI.

We also add text (Page 2, Line 36):

This approach has been applied variously to the pure ABX_3 systems, MAPbI_3 , MAPbBr_3 and FAPbI_3 , either in the form of bulk thin-film or quantum dots or nanocrystals. In all cases, it has been observed that the final decomposition product is BX_2 (e.g., PbI_2 or PbBr_2)^{18–24} or Pb ^{21,25,26}. However, the initial perovskite ABX_3 structures do not collapse directly into BX_2 , but form an intermediate phase^{18–20,23–25}.

Finally, while this study is equally applicable to bulk and nanocrystal perovskites, we note that nanocrystals offer specific advantages which have been well documented – we have added the following references accordingly:

(34) Yuan, J et al *Joule* **2020**, 4 (6), 1160–1185.

(35) Hazarika, A et al *ACS Nano* **2018**, 12 (10), 10327–10337.

(36) Zheng, K. et al *J. Phys. Chem. C* **2016**, 120 (5), 3077–3084.

Point 3: “Also, the Introduction section is very long and can be shortened”

We have removed the last paragraph and some sentences. We defer to the editor as to whether further shortening is required. We are reluctant to remove additional text as we believe this provides important context to the present work.

Point 4: When the authors draw conclusions about vacancy formation as a result of the electron dose, it is not clear how this relates to device behavior. In general, perovskites are known to be intrinsically defect tolerant. Is the defect density introduced by the electron dosage sufficient to create noticeable device performance loss? Are these defect densities and defect types a concern for the technologies expected from these semiconductors?

To clarify this, we have added the following to the conclusion:

Point defects, such as vacancies, in photoactive perovskites are generally believed to be electronically benign, due to the observed long carrier diffusion lengths and low recombination rates⁴². However, while vacancies may be electronically benign, this study shows that they

can be structurally toxic, being pivotal to ion migration and structural degradation and thereby undermining the potential of these materials for use in solar cell devices.

Even in the nominally pristine structure, occasional vacancy pairs are evident (Fig. S8). Although low density, these provide the initial 'space' essential to permit ion movement and rearrangement. Once a stimulus is applied, additional vacancy pairs can form, further facilitating ion migration and local ordering. These are the first atomic-scale steps to device hysteresis and ultimately structural degradation.

Point 5: This study could also have benefited from theoretical modeling of these defects pinpointing the location of the resulting defect states in the band structure.

As noted above, this study relates to the role of vacancies in ion migration and structural stability. We have performed DFT calculations to further understand the role of vacancies in ion migration.

We have also attempted preliminary ab-initio molecular dynamics simulations. These initial simulations indicate that a comprehensive study would take ~18 months of CPU time. This is due to the complexity of the initial aperiodic structure with ad-hoc vacancies and their subsequent diffusion into a superlattice. We have nevertheless reported the initial AIMD observations in the supplementary material.

Page 10 Line 19, add text:

We have performed density function theory (DFT) calculations to better understand the A-site cation migration and the role of vacancy in the ion migration. A-site cation migration is not realistic in a perfect perovskite structure with fully occupied ions. DFT results show that the introduction of A-site vacancies will activate the migration of remaining A-site cations (supplementary note 14). It suggests that vacancies provide driving force for the subsequent re-ordering of cations through ion migration. We have also attempted to perform ab-initio molecular dynamics simulations to model the formation of the superstructure and its diffusion on a larger supercell scale. However, conclusive results have not been reached given by the complexity of the system, as discussed in supplementary note 14.

We have added details of the calculations / simulations in (Supplementary Note 14)

Supplementary Note 14

Calculations of A-site cation migration energy barrier

The NEB results show that the migration energy barrier of FA in the structure of FAPbI₃ is higher than that of Cs in the CsPbI₃ structure. The figure below shows the migration energy barrier of FA and Cs from 1b crystallographic Wyckoff site to the neighbouring 1b FA-vacancy site and Cs-vacancy site. Compared to FA, Cs has a smaller size, which can be the main reason for the lower migration barrier energy of Cs.

Fig. S19 NEB results of FAPbI₃ and CsPbI₃.

Ab-initio molecular dynamics (AIMD) simulations were performed to enhance our understanding of the ionic diffusion mechanisms within the structures of FAPbI₃ and CsPbI₃. However, conclusive results have not been reached due to the complexity of the system. The aperiodic nature of the structure with initial ad-hoc vacancies, and its subsequent diffusion towards a superlattice, requires an exceptionally large supercell to ensure appropriate periodic boundary conditions. The computational resources required are well-beyond the scope of the present study, however, preliminary, qualitative observations from these simulations can be reported.

For each sample (FAPbI₃ and CsPbI₃ structures) three trajectories are considered, and each trajectory is simulated for a duration of 20 ps to study the dynamics of Cs atoms and FA molecules. All the calculations were done at 350 K. Based on the results, at the first 10 ps of the trajectories, Cs atoms diffused to the Cs-vacancy site resulting in a change in crystallographic symmetry. However, in none of the FAPbI₃ samples did FA exhibit diffusion within the timeframe studied. These results are consistent with the NEB results and show that, *for a given vacancy concentration*, Cs diffusion in CsPbI₃ is easier than FA diffusion in FAPbI₃. In practice, the vacancy concentration also needs to be taken into consideration. The vacancy concentration is known to affect the ionic diffusion of elements. Typically, a higher concentration of vacancies tends to lower the energy barrier for ionic diffusion. This is also observed here in the AIMD simulations. In reality, FAPbI₃ is expected to have significantly more vacancies compared with CsPbI₃. This is because FA readily decomposes into smaller molecules (such as NH₃ and CH₂N) and evaporates, facilitating vacancy formation, particularly in the presence of stimuli such as light, heat and electron beams⁹. This is consistent with our experimental observations which suggest that more FA vacancies form in FAPbI₃, compared with A-site vacancies in FA_{0.5}Cs_{0.5}PbI₃, for a given electron dose (see main manuscript). This higher vacancy concentration for FA is likely to lead to a reduction in the energy barrier for FA migration.

We have added descriptions of calculations / simulations method in the main text, method section:

DFT calculations. DFT as implemented in the Vienna Ab Initio Package (VASP)⁴³ is used to study the Cs_xFA_{1-x}PbI₃ perovskite structure. The GGA-PB⁴⁴ functional are considered for all

the calculations. To minimise the effect of periodic boundary conditions on atomic interactions, a supercell consisting of 8 unit cells with 96 atoms ($12.703 \text{ \AA} \times 12.703 \text{ \AA} \times 12.703 \text{ \AA}$) for FAPbI_3 and a supercell with 40 atoms ($12.564 \text{ \AA} \times 12.564 \text{ \AA} \times 12.564 \text{ \AA}$) for CsPbI_3 are considered for this study. A k-point of $1 \times 1 \times 1$ at the Γ -point is applied for all the calculations. To compare the migration energy barrier (NEB) of FA and Cs ions in the structure of FAPbI_3 and CsPbI_3 , nudged elastic band calculations with a force tolerance of 0.03 eV \AA^{-1} and an energy cutoff of 520 eV are carried out. To make the initial and final structures, one FA and I and similarly one Cs and I are removed from FAPbI_3 and CsPbI_3 structures ($\text{FA}_{0.875}\text{PbI}_{2.875}$ and $\text{Cs}_{0.875}\text{PbI}_{2.875}$) and DFT ground state optimisation is used to converge the initial and final migration points. The energy optimised structures are used to generate six intermediate images for NEB calculations.

Ab-initio Molecular Dynamics (MD) simulation. The samples that were used for NEB calculations are also being used for Born-Oppenheimer molecular dynamics simulations. *Ab initio* MD simulations were performed using the CP2K/Quickstep package⁴⁵. The Perdew-Burke-Ernzerhof (PBE) generalized gradient approximation (GGA)⁴⁴ was selected for the DFT exchange-correlation functional. To correct for van der Waals interactions the DFT-D3⁴⁶ method was used. Pseudopotentials of Goedecker, Teter and Hutter (GTH)⁴⁷ were employed and the DZVP-MOLOPT-SR-GTH⁴⁸ was selected as basis set. This is a Gaussian and plane-wave (GPW)⁴⁹ basis and a cutoff energy of 280 Ry was selected. A $1 \times 1 \times 1$ k-point mesh (Γ point) was used in all calculations. For *ab initio* MD simulations, the equations of motions were integrated using a velocity Verlet algorithm with a time step of 1 fs.

Reviewer #2 (Remarks to the Author):

In this study, Li and coauthors investigate the instability of Cs_{1-x}F_xPbI₃ and clarify the role of ion migration, octahedral tilt and the A-site cation via low-dose TEM. They directly image the detailed structural evolutions of Cs_{1-x}F_xPbI₃ under the electron beam irradiation including the initial random loss of vacancies, forming superstructures and inducing the octahedral tilt. Overall, this work provides insights into the atomic-scale structural mechanisms of ion loss and migration and suggests strategies for reducing ion migration and increasing structural stability. The manuscript is well organized and written. Several issues need to be addressed before publication, which are detailed below.

Point 1. My first and foremost concern is the influence of post-filtering. In the raw image, there is no additional reflections while, in the filtered images, additional reflections and the intensity modulation are observed. Why these filters can produce additional reflections?

We understand that it is important to ensure image processing does not generate artefacts.

Firstly, to assure the reader that the vacancies are real, we prove that vacancies and their ordering are evident in the original, unfiltered raw STEM-ADF images, as follows:

Supplementary Note 3 – 2nd paragraph and Fig S2:

It is evident that the structure imaged in Fig. S2A (unfiltered raw data of image Fig. 1E) is not consistent with the established cubic structure of FAPbI₃, despite the use of the lowest achievable dose conditions for the STEM-ADF image. STEM-ADF images are composition-sensitive so the existence of A-site vacancies can be observed directly from the image as indicated by the lower image intensity within the dashed red squares. Furthermore, where they occur, these vacancies tend to be ordered, as seen from the image and corresponding intensity line profile in Fig. S2C.

Secondly, we further demonstrate that reflections associated with the vacancies in the Fourier Transform of the corresponding unfiltered raw image were *not* visible and we explain why this is.

Supplementary Note 3 – opening paragraph:

We would like to make an important point here regarding the identification of the electron dose at which damage occurs. This is often done through the identification of additional reflections in the Fourier transform (FT) of an image that are inconsistent with the pristine structure. We emphasise here that subtle, local damage can occur on a scale that is too small and aperiodic to be detectable in the Fourier transform of the image, hence Fourier transforms may fail to detect the early stages of damage and are not a reliable measure of the lowest dose limit at which electron beam damage might occur. The current work is a case in point, which we highlight in this supplementary note.

Supplementary Note 3 – 3rd paragraph:

The observed vacancy superstructure has a lower symmetry presumed to have been induced by exposure to the electron beam. However, the image intensity modulations generated by this superstructure are **too subtle and too localised** to be detectable as additional reflections in the corresponding Fourier transform (FT) of the image, which instead appears to be consistent with the pristine cubic structure (Fig. S3A). ~~That is, the intensity modulation is too subtle and too localised to generate visible additional reflections in the FT (that would have~~

been forbidden for the pristine cubic structure, Fig. S3(B,C)). To expand on this point, the signal-to-noise ratio (SNR) of the first STEM-ADF image (Fig. S2A) is extremely low (because it was deliberately taken at the lowest possible STEM-ADF electron dose conditions). In addition, due to the low electron dose used in this initial image, minimal damage has been done, so the superstructure is only present in a few local regions, insufficient to generate additional 'narrow-frequency-band' reflections above the broad-band noise in the FT.

Fig. S2 $\text{Cs}_{0.5}\text{FA}_{0.5}\text{PbI}_3$ showing initial, localised vacancies detectable in the raw data image, presumed to be due to initial beam damage. (A) Raw STEM-ADF image of $\text{Cs}_{0.5}\text{FA}_{0.5}\text{PbI}_3$ taken at $44 \text{ e}/\text{\AA}^2$. (B) Enlarged image of region marked in (A). (C) Intensity line profile measured from the green rectangle. Positions in the plot corresponding to the A-site atomic positions are highlighted by squares (blue for higher intensity, red for lower intensity and hence lower occupancy).

Fig. S3 $\text{Cs}_{0.5}\text{FA}_{0.5}\text{PbI}_3$ - Fourier transform of lowest dose, raw STEM-ADF image (Fig S2) – the signal from the local vacancies is not detectable above the noise in the FT. (A) FT of the STEM-ADF in Fig. S1A. (B) Intensity line profile from the region highlighted by red shows no additional $1/2, 1/2, 0_c$ reflections that would be associated with the ordered vacancies. (C) Intensity line profile from the region highlighted by blue shows no additional $3/2, 1/2, 0_c$ reflections that would be associated with the ordered vacancies.

Finally, we have summarised this by adding to the main manuscript:
Page 3, Line 41:

We also confirm that these vacancies and ordering are evident in the lowest dose raw data images and are not introduced by the post-filtering process (supplementary note 3, Fig. S2).

Point 2. Besides, the ABSF and Winer filter images seem to match better with the raw image and no superstructure reflections are observed in the corresponding FFT pattern. However, the vacancies there seem not ordered. Thus, how to identify that the observed intensity-modulation is the pristine feature in the raw images rather than induced by the Bragg filter and butterworth filter?

We explain this in the following modified text and figures in (Supplementary Note 4):

To reduce the effect of noise, we applied different filters, all of which enhance the contrast from the $\sqrt{2} \times \sqrt{2}$ superstructures in the image (Fig. S4B-D). It is important that any filtered images should be compared with raw images to make sure that post-filtering does not create features/artefacts that are not present in the original raw images. In Fig. S5, the filtered images are compared with the raw image. The combination of Bragg-Butterworth filter in Fig. S5B shows A-site vacancies and their positions and ordering are the same as revealed in the raw image in Fig. S5A (same figure as Fig S2A). Furthermore, the filtering helps to visualise the halide columns. The Bragg-Butterworth filter has also been utilized in the literature for the

analysis of low-dose STEM-ADF images of FAPbI_3 , and it has also not been found there to show structural artefacts. All structures (e.g. vacancies, ordering and octahedral tilt) were also evident in the raw images⁶.

We also applied other commonly used image filters, such as the ABSF filter and the Winer filter to the low-dose STEM-ADF image (Fig S5A), see Fig. S5(C, D). They also show the same A-site vacancies and ordering that exist in the raw image.

The only Fourier transform to show additional reflections associated with the vacancy ordering (which is unequivocally evident in all the images) is the FT of the Bragg-Butterworth filtered image (Fig S4B). This is to be expected, as the Bragg-Butterworth filter is the most efficient in terms of the balance between de-noising and the retention of high frequency structural information.

Fig. S4 Post-filtering of an example lowest dose STEM-ADF image taken from $\text{Cs}_{0.5}\text{FA}_{0.5}\text{PbI}_3$, with a total dose at $44 \text{ e}/\text{\AA}^2$. (A) Raw STEM-ADF image. (B) Image filtered by a combination of Bragg filter and Butterworth filter. (C) Image filtered by a Winer filter. (D) Image filtered by an average background subtraction filter (ABSF). Corresponding FTs are given below the overview images.

Fig. S5 Post-filtering of an example low dose STEM-ADF image taken from $\text{Cs}_{0.5}\text{FA}_{0.5}\text{PbI}_3$, with a total dose at $44 \text{ e}/\text{\AA}^2$. Enlarged images of marked regions in Fig. S4. (A) Raw image, (B) Bragg-Butterworth filtered image, (C) Winer filtered image, (D) ABSF filtered image. All the filtered images exhibit vacancies and ordering in the same location as the original, raw unfiltered image.

Point 3. I suggest the author analyzes the intensity modulation from the images containing additional reflections in the raw images (just like STEM image in Fig. 4a).

In the original manuscript, all the images show clear additional reflections in their Fourier transforms (e.g., Fig. 2, Fig. 3, Fig. 4), except for the raw data image (Fig. S3) corresponding to Fig 1. We have discussed the reasons for Fig. 1 in our response to Points 1 and 2.

We appreciate the reviewer's suggestion but think it is important to retain Fig 1/FigS3. As discussed in our response to points 1 and 2, the vacancies evident in these images are real and are not an artefact. Furthermore, this data provides an important example as to why the research community should NOT use additional reflections in FTs to estimate the minimum dose before damage occurs. As discussed in our response to point 1, damage may occur well before it is apparent in the FT. Superstructure reflections in FTs have long been used as a measure of the beam damage threshold. In Supplementary note 3 (including Fig 1/Fig S3), we show this can be incorrect and can lead to an overestimate of the minimum acceptable electron dose.

4. Please provide the raw image and methods of filtering for each image.

We thank the reviewer for this suggestion. We have added an Appendix section at the end of Supporting Information and include all raw images and the corresponding filtered images, as used in the manuscript. References describing the method for applying each type of filter are also provided.

Raw images in its original file formats will also be provided to the publisher at submission.

Appendix

Raw images and filtering for each image

Raw/Filtered Low-dose STEM-ADF images in Fig. 1, Fig. 3 and Fig. 4 from the main text are summarized in the appendix. A Bragg-Butterworth filter^{10,11} has been applied to these images as detailed in Fig. S4B.

Figure 1

Figure 2

Figure 3

Figure 4

5. How to identify the type of atoms in phase-contrast HRTEM images?

We have added Supplementary Note 6 to identify the type of atoms:

Supplementary Note 7

Determination of atom types in phase-contrast HR-TEM images

Unlike STEM-HAADF images where the intensity of atomic columns can be related directly to the type and number of atoms in the column, contrast in phase-contrast HR-TEM images can be significantly affected by specimen thickness and imaging conditions, such as defocus and lens aberrations. In the HR-TEM images in Fig. S9, three levels of intensity, A to C, are observed at atomic column positions. We identify the atom types associated with these 3 intensity levels and positions, as follows:

Firstly, the intensity B can be determined as corresponding to the corners of the octahedra due to their unique arrangement relative to A and C. The corners contain iodine only, so we allocate the B-intensity to pure iodine columns. Then, A and C should correspond to columns contain FA^+ or alternating $\text{Pb}^{2+}/\text{I}^-$. There are two observations that indicate that the C-type column corresponds to the FA^+ and the A-type column corresponds to the $\text{Pb}^{2+}/\text{I}^-$, as follows:

- (1) Column shape. The shape of $\text{Pb}^{2+}/\text{I}^-$ columns in this projection is expected to be spherical while the FA^+ molecule is non-spherical and is also expected to rotate in arbitrary orientations at this temperature (Fig. S9C). In the HR-TEM image, the A-type intensity distribution is consistently observed to have a spherical shape consistent with $\text{Pb}^{2+}/\text{I}^-$ (or a slight elliptical shape due to a slight tilt away from the zone axis), however, the C-type intensity distribution is much more variable in shape consistent with FA^+ .
- (2) Vacancy. From the STEM-ADF image (Fig. 1), where the atom types can be easily determined, we observed the occupancy of the FA^+ columns varies significantly, while the occupancy of the $\text{Pb}^{2+}/\text{I}^-$ columns is much more stable (at least at the low doses used for this image). In the HR-TEM image, the A-type intensity shows a minor intensity variation while the C-type intensity shows significant variations. This is again consistent with the A position corresponding to $\text{Pb}^{2+}/\text{I}^-$ columns and C corresponding to FA^+ columns.

Fig. S9 Phase-contrast HR-TEM image of FAPbI₃ – identification of atom types. (A) Same image as Fig. 2D at 245 e/Å². (B) Enlarged image of the region marked in (A). (C) structure model of FAPbI₃ unit cell with FA⁺ perpendicular to the view direction. (D) Intensity line profile across A-type intensity maxima. (E) Intensity line profile across C-type intensity maxima.

6. Could the author provide a more quantitative analysis on estimating the vacancy content based on the intensity of the STEM images for Fig. 1?

We have added Supplementary Note 5 to provide further quantitative analysis of Fig. 1. However, we note that it is not feasible to determine the *absolute* composition quantitatively from the image intensity for these low dose images due to their extremely low signal to noise:

Supplementary Note 5

Quantitative estimation of A-site vacancies and A-site/halide vacancy pairs

A more quantitative analysis of the A-site column intensity in Figure 1 is shown in Fig. S6, where the intensity in a Voronoi cell around each atomic column position has been integrated. This enhances the intensity variation evident at the A-site columns, with a lower intensity at A-site columns attributed to vacancies in that column. The ordering of A-site vacancies results in a checkered pattern in the integrated A-site intensity map. In this initial image, the checkered pattern is only observed in some areas, which is likely because, at this low dose, there is limited damage so relatively few A-site vacancies have been generated.

Fig. S6 Quantitative analyses of A-site intensity from low-dose STEM-ADF image of Cs_{0.5}FA_{0.5}PbI₃. (A) Filtered image (the same as that in Fig. 1E), (B) map of integrated intensity of A-site columns.

As discussed in the main manuscript, vacancy and vacancy-ordered superstructures were observed for both A-sites and halide sites (Fig 1). The position of vacancy containing columns is highlighted in Fig. S7. This suggests that cation vacancies and anion vacancies occur in pairs and form correlated superstructures. The number of cation and anion vacancies are expected to be identical to ensure charge neutrality and this appears to be the case from the image, in so far as it can be quantified.

Fig. S7 Analysis of A-site vacancies and I-site vacancies from low-dose STEM-ADF image of $\text{Cs}_{0.5}\text{FA}_{0.5}\text{PbI}_3$ (the same as Fig. 1F).

7. Fig. 1b and 1f, the region of lineprofiles is not clearly indicated.

8. FFT patterns in Fig. 1 is too small and should be removed to SI. Moreover, FFT patterns in Fig.2-5 can be further cropped to make the spots clearer.

We appreciate the reviewer's suggestions. We have endeavoured to clarify Fig. 1 by removing the FT patterns (which are now shown in Fig. S3) and further highlight the regions used for the line profiles. We have also cropped and enlarged the FT patterns in Fig. 2-5, to make the details easier to see.

Reviewer #3 (Remarks to the Author):

The structural degradation of organic-inorganic hybrid perovskite solar cells is a major obstacle in their commercialization. To investigate the degradation mechanism, the authors used transmission electron microscopy-based techniques to explore the effects of ion migration, octahedral tilt, and the A-site cation on $\text{Cs}_{1-x}\text{FA}_x\text{PbI}_3$ instability. The authors made several notable novel findings, including the atomic-scale dynamic observation of superstructure formation and distinct degradation pathways for FAPbI_3 and $\text{Cs}_{1-x}\text{FA}_x\text{PbI}_3$, which significantly advance the understanding of its degradation mechanism. However, there are some comments that the authors should take into account before publication:

1. Although the authors mainly focused on investigating the initial intermediate phases, it is essential to consider the evidence for the transition from the intermediate phase to the final PbI_2 phase, which is well known to result from the decomposition of ABi_3 .

Indeed, this paper focusses on revealing the very first, atom-by-atom structural changes. The final decomposition product has been well studied and reported for almost all perovskite solar cell systems to be PbI_2 , so we did not focus on this here. To address this, we have added in (Supplementary Note 12):

Once the electron dose exceeds the dose threshold that can be sustained by the perovskite framework, the perovskite structure further decomposes into PbI_2 (Fig S17), consistent with previous TEM studies of this system⁸ and other perovskite solar cell systems^{6,7}. We also notice that the perovskite phase and PbI_2 phase can co-exist in the mixed cation $\text{Cs}_{0.5}\text{FA}_{0.5}\text{PbI}_3$. This could be due to the presence of ad-hoc vacancies in the pre-damaged, pristine structure, so some regions have a 'head-start' towards decomposition over others. Another hypothesis is that there is an inhomogeneous distribution of Cs^+ and FA^+ . Given FA^+ vacancies are expected to occur more easily⁷, at the electron dose that FA^+ rich regions have decomposed completely into PbI_2 , some Cs^+ rich regions still retain the perovskite structures.

Fig. S17 Decomposition $\text{Cs}_{0.5}\text{FA}_{0.5}\text{PbI}_3$ into final PbI_2 phase. (A) STEM-ADF image; (B) FT of the region marked in red in (A); (C) FT of the region marked in black in (A).

2. The authors observed an abnormal phenomenon of vacancy-induced superstructure formation, but the proposed discussion of ordered cation diffusion needs proper simulations for better understanding.

We have performed DFT calculations to further understand the role of vacancies in ion migration.

We have also attempted preliminary ab-initio molecular dynamics simulations. These initial simulations indicate that a comprehensive study would take ~18 months of CPU time. This is

due to the complexity of the initial aperiodic structure with ad-hoc vacancies and their subsequent diffusion into a superlattice. We have nevertheless reported the initial AIMD observations in the supplementary material.

Page 10 Line 19, add text:

We have performed density function theory (DFT) calculations to better understand the A-site cation migration and the role of vacancy in the ion migration. A-site cation migration is not realistic in a perfect perovskite structure with fully occupied ions. DFT results show that the introduction of A-site vacancies will activate the migration of remaining A-site cations (supplementary note 14). It suggests that vacancies provide driving force for the subsequent re-ordering of cations through ion migration. We have also attempted to perform an-initio molecular dynamics simulations to model the formation of the superstructure and its diffusion on a larger supercell scale. However, conclusive results have not been reached given by the complexity of the system, as discussed in supplementary note 14.

We have added details of calculations / simulations in (Supplementary Note 14):

Supplementary Note 14

Calculations of A-site cation migration energy barrier

The NEB results show that the migration energy barrier of FA in the structure of FAPbI₃ is higher than that of Cs in the CsPbI₃ structure. The figure below shows the migration energy barrier of FA and Cs from 1b crystallographic Wyckoff site to the neighbouring 1b FA-vacancy site and Cs-vacancy site. Compared to FA, Cs has a smaller size, which can be the main reason for the lower migration barrier energy of Cs.

Fig. S19 NEB results of FAPbI₃ and CsPbI₃.

Ab-initio molecular dynamics (AIMD) simulations were performed to enhance our understanding of the ionic diffusion mechanisms within the structures of FAPbI₃ and CsPbI₃. However, conclusive results have not been reached due to the complexity of the system. The aperiodic nature of the structure with initial ad-hoc vacancies, and its subsequent diffusion towards a superlattice, requires an exceptionally large supercell to ensure appropriate periodic

boundary conditions. The computational resources required are well-beyond the scope of the present study, however, preliminary, qualitative observations from these simulations can be reported.

For each sample (FAPbI₃ and CsPbI₃ structures) three trajectories are considered, and each trajectory is simulated for a duration of 20 ps to study the dynamics of Cs atoms and FA molecules. All the calculations were done at 350 K. Based on the results, at the first 10 ps of the trajectories, Cs atoms diffused to the Cs-vacancy site resulting in a change in crystallographic symmetry. However, in none of the FAPbI₃ samples did FA exhibit diffusion within the timeframe studied. These results are consistent with the NEB results and show that, *for a given vacancy concentration*, Cs diffusion in CsPbI₃ is easier than FA diffusion in FAPbI₃. In practice, the vacancy concentration also needs to be taken into consideration. The vacancy concentration is known to affect the ionic diffusion of elements. Typically, a higher concentration of vacancies tends to lower the energy barrier for ionic diffusion. This is also observed here in the AIMD simulations. In reality, FAPbI₃ is expected to have significantly more vacancies compared with CsPbI₃. This is because FA readily decomposes into smaller molecules (such as NH₃ and CH₂N) and evaporates, facilitating vacancy formation, particularly in the presence of stimuli such as light, heat and electron beams⁹. This is consistent with our experimental observations which suggest that more FA vacancies form in FAPbI₃, compared with A-site vacancies in FA_{0.5}Cs_{0.5}PbI₃, for a given electron dose (see main manuscript). This higher vacancy concentration for FA is likely to lead to a reduction in the energy barrier for FA migration.

We have added descriptions of calculations / simulations method in the main text, method section:

DFT calculations. DFT as implemented in the Vienna Ab Initio Package (VASP)⁴³ is used to study the Cs_xFA_{1-x}PbI₃ perovskite structure. The GGA-PB⁴⁴ functional are considered for all the calculations. To minimise the effect of periodic boundary conditions on atomic interactions, a supercell consisting of 8 unit cells with 96 atoms (12.703 Å × 12.703 Å × 12.703 Å) for FAPbI₃ and a supercell with 40 atoms (12.564 Å × 12.564 Å × 12.564 Å) for CsPbI₃ are considered for this study. A k-point of 1 × 1 × 1 at the Γ -point is applied for all the calculations. To compare the migration energy barrier (NEB) of FA and Cs ions in the structure of FAPbI₃ and CsPbI₃, nudge elastic band calculations with a force tolerance of 0.03 eV Å⁻¹ and an energy cutoff of 520 eV are carried out. To make the initial and final structures, one FA and I and similarly one Cs and I are removed from FAPbI₃ and CsPbI₃ structures (FA_{0.875}PbI_{2.875} and Cs_{0.875}PbI_{2.875}) and DFT ground state optimisation is used to converge the initial and final migration points. The energy optimised structures are used to generate six intimate images for NEB calculations.

Ab-initio Molecular Dynamics (MD) simulation. The samples that were used for NEB calculations are also being used for Born-Oppenheimer molecular dynamics simulations. *Ab initio* MD simulations were performed using the CP2K/Quickstep package⁴⁵. The Perdew-Burke-Ernzerhof (PBE) generalized gradient approximation (GGA)⁴⁴ was selected for the DFT exchange-correlation functional. To correct for van der Waals interactions the DFT-D3⁴⁶ method was used. Pseudopotentials of Goedecker, Teter and Hutter (GTH)⁴⁷ were employed and the DZVP-MOLOPT-SR-GTH⁴⁸ was selected as basis set. This is a Gaussian and plane-wave (GPW)⁴⁹ basis and a cutoff energy of 280 Ry was selected. A 1×1×1 k-point mesh (Γ point) was used in all calculations. For *ab initio* MD simulations, the equations of motions were integrated using a velocity Verlet algorithm with a time step of 1 fs.

3. The authors should clarify whether the random vacancies observed in Figure 1 are due to beam irradiation or sample preparation.

We have added text, as follows:

Page 5, Line 31:

However, the number of vacancies that were observed at this stage (HR-TEM @ $7.8 \text{ e}/\text{\AA}^2$) is much less than in the first acquired STEM-ADF images taken with $44 \text{ e}/\text{\AA}^2$ (Fig. 1). This suggests at least some of the vacancies observed in the first STEM-ADF images (Fig. 1) were induced by the electron beam, even though the STEM-ADF image was taken with the lowest achievable dose for STEM-ADF. We cannot know whether the vacancies incurred in the HR-TEM (taken at $7.8 \text{ e}/\text{\AA}^2$) are intrinsic to the specimen or induced by the electron beam. We hypothesise that it is likely both are true.

4. The authors claim that charge neutralization plays a crucial role in cation vacancy formation, which suggests that the number of vacancies for A cations and I- should be similar. It would be helpful if the authors could provide quantitative analysis from the images to support this claim.

We have added Supplementary Note 5 to provide further quantitative analysis of Fig. 1 which shows a reduced intensity at a given A-site column is accompanied by reduced intensity at the adjacent halide column. However, a more quantitative measurement of the absolute number of vacancies and vacancy positions in three-dimensions is not feasible for these low dose images.

Supplementary Note 5

Quantitative estimation of A-site vacancies and A-halide vacancy pairs

As discussed in the main manuscript, vacancy and vacancy-ordered superstructures were observed for both A-sites and halide sites (Fig 1). The position of vacancy containing columns are highlighted in Fig. S7. This suggests that cation vacancies and anion vacancies occur in pairs and form correlated superstructures. The number of cation and anion vacancies are expected to be identical to ensure charge neutrality and this appears to be the case from the image, in so far as it can be quantified.

Fig. S7 Analysis of A-site vacancies and I-site vacancies from low-dose STEM-ADF image of $\text{Cs}_{0.5}\text{FA}_{0.5}\text{PbI}_3$ (the same as Fig. 1F).

5. The authors named intermediate as phase 1 and phase 2, and it would be useful to discuss the corresponding non-stoichiometric chemical formula based on the observations.

We thank the reviewer for this helpful suggestion. We have added the corresponding non-stoichiometric chemical formula to Fig. 6 and refer to it in the corresponding text in the manuscript, as follows:

Fig. 6 schematics of the observed ion migration mechanisms and associated phase changes of $\text{Cs}_{1-x}\text{FA}_x\text{PbI}_3$ under the electron beam. (A) FAPbI_3 . (B) $\text{Cs}_{0.5}\text{FA}_{0.5}\text{PbI}_3$.

Added text in the discussion section:

Page 10, Line 8:

The initial loss of ions would result in a nominal formula of $(\text{FA}_{(1-x)}\text{PbI}_{(3-x)})$, $x=0-0.5$) or $(\text{Cs}_{0.5}\text{FA}_{(0.5-x)}\text{PbI}_{(3-x)})$, $x=0-0.25$).

Page 10, Line 21:

Our third observation is that once vacancies have formed a $\sqrt{2} \times \sqrt{2}$ superstructure, a relatively stable intermediate phase (with nominal formula of $\text{FA}_{0.5}\text{PbI}_{2.5}$ or $\text{Cs}_{0.5}\text{FA}_{0.25}\text{PbI}_{2.75}$) is established in which ion migration is substantially impeded, despite the presence of a significant number of FA^+/I^- vacancies.

Page 10, Line 32:

At this stage the nominal formula corresponds to $(\text{FA}_{(0.5-x)}\text{PbI}_{(2.5-x)})$, $x=0-0.5$) or $(\text{Cs}_{0.5}\text{FA}_{(0.25-x)}\text{PbI}_{(2.75-x)})$, $x=0-0.25$) before completely decomposing into PbI_2 (Fig. S17).

REVIEWERS' COMMENTS

Reviewer #1 (Remarks to the Author):

Authors have included explanations in response to the points I had raised, for which they are thanked. While the overall TEM microscopy and analysis presented is unique, I continue to stand by my decision that these results are specific to colloidal quantum dot systems, and these systems differ significantly from bulk perovskite films in their degradation modes. I therefore do not think that these findings will be generic to researchers working to improve stability of perovskite thin films and solar cells. I defer to the editor's judgment on this matter.

Reviewer #2 (Remarks to the Author):

The authors have well solved my concern, I recommend the revised version of this manuscript for publication.

Reviewer #3 (Remarks to the Author):

The authors have addressed my concerns and now I recommend the publication of this work in nat commun.

RESPONSE TO REVIEWERS' COMMENTS

Response to Reviewer #1

Manuscript ID: **NCOMMS-23-20206A**

Title: "Instability of Cs_{1-x}F_xPbI₃ – the role of ion migration, octahedral tilt and the A-site cation"

Author(s): Li, Weilun; Hao, Mengmeng; Wang, Ardeshir Baktash, Lianzhou; Etheridge, Joanne

We thank the reviewers for their time and helpful comments. We have endeavoured to address Reviewer #1's additional comment in the revised manuscript, as detailed below. (Reviewer #1's comments in **black bold**, our reply in **blue**, changes to the manuscript in **red**. Page and line number refers to the original manuscript)

Reviewer #1 (Remarks to the Author):

Authors have included explanations in response to the points I had raised, for which they are thanked. While the overall TEM microscopy and analysis presented is unique, I continue to stand by my decision that these results are specific to colloidal quantum dot systems, and these systems differ significantly from bulk perovskite films in their degradation modes. I therefore do not think that these findings will be generic to researchers working to improve stability of perovskite thin films and solar cells. I defer to the editor's judgment on this matter.

We have endeavoured to clarify and further emphasise this from two aspects.

Firstly, structural degradation in other photoactive perovskites has been reported to be the same, irrespective of the specimen preparation method (e.g. vacuum deposited FAPbI₃ and MAPbI₃ thin films (*Science* 370.6516 (2020): eabb5940.), solution processed MAPbI₃ thin films (*Adv. Materials* 30.25 (2018): 1800629, and a variety of other processing methods *Adv. Materials* (2023): 2211207, *Nature Comm.*12.1 (2021): 5516 and *Nature Comm.* 9.1 (2018): 4807.).

Added, Page 2, Line 37:

"This approach has been applied variously to the pure ABX₃ systems, MAPbI₃, MAPbBr₃ and FAPbI₃, either in the form of bulk thin-film or quantum dots or nanocrystals. **The structural response in these systems has been shown to be the same irrespective of whether they are in quantum dot or bulk form¹⁸⁻²⁰.**"

Secondly, photo-activate perovskites in the form of quantum dots have some distinctive features compared with their thin-film (or bulk) counterparts, such as additional bandgap tuning via the quantum-confinement effect, multiexciton generation, and improved phase stability. These features offer their own exciting applications.

Added, Page 3, Line 12:

"We also note that quantum dots have their own exciting applications in photo-active devices^{37,38}."